# Biologically indeterminate yet ordered promiscuous gene expression in single medullary thymic epithelial cells

Fatima Dhalla[1],[†] [iD], Jeanette Baran-Gale[2],[†] [iD], Stefano Maio[1] [iD], Lia Chappell[3] [iD], Georg A Holl
änder[1],[*] [iD] &
Chris P Ponting[2],[3],[**] [iD]

## Abstract

**To induce central T-cell tolerance, medullary thymic epithelial cells (mTEC) collectively express most protein-coding genes, thereby presenting an extensive library of tissue-restricted antigens (TRAs). To resolve mTEC diversity and whether promiscuous gene expression (PGE) is stochastic or coordinated, we sequenced transcriptomes of 6,894 single mTEC, enriching for 1,795 rare cells expressing either of two TRAs, TSPAN8 or GP2. Transcriptional heterogeneity allowed partitioning of mTEC into 15 reproducible subpopulations representing distinct maturational trajectories, stages and subtypes, including novel mTEC subsets, such as chemokine-expressing and ciliated TEC, which warrant further characterisation. Unexpectedly, 50 modules of genes were robustly defined each showing patterns of co-expression within individual cells, which were mainly not explicable by chromosomal location, biological pathway or tissue specificity. Further, TSPAN8[+] and GP2[+] mTEC were randomly dispersed within thymic medullary islands. Consequently, these data support observations that PGE exhibits ordered co-expression, although mechanisms underlying this instruction remain biologically indeterminate. Ordered co-expression and random spatial distribution of a diverse range of TRAs likely enhance their presentation and encounter with passing thymocytes, while maintaining mTEC identity.**

**Keywords** autoimmune regulator; central T-cell tolerance; medullary thymic epithelial cells; promiscuous gene expression; thymus
**Subject Categories** Chromatin, Transcription & Genomics; Immunology
**The EMBO Journal (2020) 39: e101828**

## Introduction

Types of differentiated cells are distinguished by their restricted expression of transcription factors, upstream regulator proteins and downstream target genes. If recapitulated out of context in other cell types, transcriptional programmes can induce the reprogramming of one mature somatic cell type into another (i.e. transdifferentiation) or trigger oncogenesis (Todd & Wong, 1999). Thymic epithelial cells (TEC), the major stromal cell constituent of the thymus (Barthlott et al, 2006; Takahama, 2006; Abramson & Anderson, 2017), express almost the entire protein-coding genome (Sansom et al, 2014; Brennecke et al, 2015) and thus harbour an increased risk of transdifferentiation and consequently losing cellular identity. This capacity includes the competence to transcribe tissue-restricted genes (TRGs) whose expression in the periphery is normally limited to a single or small subset of tissues and genes whose expression is temporally or developmentally controlled or is sex-specific (Derbinski et al, 2001; Kyewski & Klein, 2006). This exhaustive transcriptional programme, termed promiscuous gene expression (PGE), provides a molecular mirror of the body's self-antigens within TEC for the purposes of central T-cell tolerance induction.

T cells that are unable to discriminate correctly between self- and non-self proteins risk provoking autoimmune disease. Therefore, during their intrathymic development, T cells are subjected to stringent selection processes mediated by recognition of self-peptide:: MHC complexes presented on the cell surface of TEC. TEC can be broadly categorised into cortical (c−) and medullary (m−) lineages based on their structure, anatomical location, molecular characteristics and functions (Rodewald, 2008; Vaidya et al, 2016). Studies examining TEC development and diversity have further revealed considerable heterogeneity within the mTEC compartment reflecting both maturationally and functionally distinct mTEC subpopulations (Nishikawa et al, 2010; Metzger et al, 2013; Bornstein et al, 2018; Miragaia et al, 2018). During intrathymic selection, cTEC positively select thymocytes that express T-cell receptors (TCRs) capable of

1 Weatherall Institute of Molecular Medicine, University of Oxford, Oxford, UK
2 MRC Human Genetics Unit, MRC IGMM, The University of Edinburgh, Edinburgh, UK
3 Wellcome Sanger Institute, Hinxton, UK
 *Corresponding author. Tel: +44 186 523 4238; E-mail: georg.hollander@paediatrics.ox.ac.uk
 **Corresponding author. Tel: +44 131 651 8500; E-mail: chris.ponting@igmm.ed.ac.uk
 †These authors contributed equally to this work

recognising peptide:MHC complexes. Following positive selection, cTEC and then mTEC remove potentially autoreactive T cells bearing high-affinity TCRs for self-antigens via negative selection and mTEC additionally redirect those with intermediate affinity to a regulatory T-cell fate (Takahama, 2006; Klein *et al*, 2009; Sansom *et al*, 2014). Only 1–3% of thymocytes successfully fulfil the stringent criteria of thymic selection and exit to the periphery (Hogquist & Jameson, 2014; Klein *et al*, 2014).

Promiscuous gene expression is partly under the control of the autoimmune regulator (AIRE), a transcriptional facilitator expressed in a subset of mature mTEC where it plays a role in the expression of just under 4,000 genes (Sansom *et al*, 2014). Around 533 of these are entirely dependent on AIRE for their expression (AIRE-dependent), and the expression of the remaining 3,260 is enhanced in the presence of AIRE (AIRE-enhanced) (Sansom *et al*, 2014). AIRE-independent mechanisms control the promiscuous expression of 3,947 TRGs (Sansom *et al*, 2014).

Despite TEC expressing at the population level almost all protein-coding genes, TRG expression at single-cell resolution is heterogeneous, with individual mature mTEC expressing only 1–3% of TRGs at one time (Derbinski *et al*, 2008; Villaseñor *et al*, 2008; Sansom *et al*, 2014; Brennecke *et al*, 2015; Meredith *et al*, 2015). A possible outcome of this mosaic expression pattern is the attainment of sufficiently high densities of particular self-antigen::MHC complexes on the cell surface of TEC to elicit tolerogenic conditions within self-reactive thymocytes (Villaseñor *et al*, 2008).

Four molecular processes, not all mutually exclusive, could explain the heterogeneity of TRG expression within single mTEC (Fig 1). Type 1: TRG expression within single mTEC is entirely stochastic. Type 2: Different maturational stages or classes of mTEC activate TRG expression to different extents (with respect to breadth and/or level of gene expression) or, alternatively, activate different TRG subsets. Type 3: A programme of TRG co-expression otherwise evident in peripheral tissues is activated. In this scenario, TRGs whose expression is restricted to a particular tissue (e.g. heart) would be transcribed concurrently as a result of a transcriptional activation programme from that peripheral tissue being co-opted. This mechanism would arguably be the most hazardous with regard to the risk of transdifferentiation. As a potential example of this type, the recently identified tuft cell-like mTEC express a programme of genes contributing to the canonical taste transduction pathway (Bornstein *et al*, 2018; Miller *et al*, 2018). Type 4: TRGs are expressed co-ordinately owing to their physical co-location by one of two mechanisms: the loci are (i) positioned contiguously on the same chromosome or (ii) distantly located but are in close vicinity due to chromatin looping.

To resolve which of these four mechanisms contribute to the thymic representation of self-antigens and to further resolve the extent of mTEC heterogeneity, we undertook large-scale single-cell RNA-sequencing of mTEC. Understanding how PGE is regulated within single TEC is essential for understanding the mechanisms by which these cells achieve their uniquely broad transcriptional programme without subverting their cellular identity. Previous studies have been limited by low cell numbers, with only several hundred mTEC analysed. We, therefore, undertook a study of the transcriptomes of thousands of single mTEC intending to resolve the existence and degree of PGE co-expression within them. Our selection of mTEC was both broad and narrow. The broad range of

mTEC were unselected with respect to tissue-restricted antigen (TRA) expression; the narrow range contained two sets of rare mTEC that expressed either tetraspanin 8 (TSPAN8) or glycoprotein 2 (GP2), two AIRE-regulated TRAs (Sansom *et al*, 2014; Rattay *et al*, 2016). TSPAN8 is expressed in the gastrointestinal tract and several carcinomas (Agaësse *et al*, 2017; Zhu *et al*, 2017; Zhao *et al*, 2018), and GP2 is expressed in the pancreas and gastrointestinal tract (Ohno & Hase, 2010; Cogger *et al*, 2017); loss of tolerance to GP2 is associated with Crohn's disease and primary sclerosing cholangitis (Werner *et al*, 2013; Tornai *et al*, 2018).

# Results

## Large-scale single-cell RNA-sequencing data from FACS sorted mTEC

We chose to analyse the transcriptomes of single mTEC that were unselected or that promiscuously expressed either TSPAN8 or GP2 (Sansom *et al*, 2014; Rattay *et al*, 2016) with the aim of adding statistical power to co-expression analyses. Expression of either TRA is enhanced in mTEC by the presence of AIRE and can be detected on the cell surface by flow cytometry. TSPAN8$^+$ mTEC constitute approximately 7% and GP2$^+$ mTEC about 2% of total mTEC (Fig 2A and Appendix Fig S1) and each continue to actively transcribe their respective genes (Fig 2B). Approximately 1% of all mTEC co-express TSPAN8 and GP2 proteins (Fig 2A).

mTEC sequencing libraries were derived from 15 individual female mice across three independent experiments. To examine strain-specific patterns in mTEC gene expression, C57BL/6 ($n = 9$) and BALB/c ($n = 2$) mice were investigated as well as their F1 cross, C57BL/6 × BALB/c ($n = 4$).

The transcriptomes of 6,894 single mTEC were analysed including 794 TSPAN8$^+$, 935 TSPAN8$^-$, 1,001 GP2$^+$, 395 GP2$^-$ and 3,769 unselected mTEC (Fig 2C). Together, these cells showed expression of 22,819 genes encoded across all chromosomes (except Y, as expected), including 19,091/21,663 or 88% of protein-coding genes. We further categorised these transcripts according to their dependence on AIRE (as defined in Sansom *et al*, 2014) and observed the expression of 89% of AIRE-dependent ($N = 477$), 98% of AIRE-enhanced ($N = 3,210$) and 94% of AIRE-independent TRGs ($N = 3,720$; Appendix Fig S2A). Therefore, the single cells largely recapitulated expression observed in TEC population-level analyses (Sansom *et al*, 2014). A median of 1,830 genes was detected per cell of which 50 (median) were AIRE-regulated TRGs (Appendix Fig S2C and D).

## Cell subpopulations were robustly identified across independent datasets

The transcriptomes of the 6,894 mTEC were projected into a reduced dimensional space resulting in a large, contiguous, central "body" of cells surrounded by several "satellite" clusters (Fig 2D). Within the central body, the majority of mTEC fell along a manifold characterised by a transition from predominantly TSPAN8$^-$ or GP2$^-$ mTEC at the lower right pole, to predominantly TSPAN8$^+$ or GP2$^+$ mTEC at the upper left pole (Figs 2D and 3A–D). TSPAN8$^+$ and GP2$^+$ mTEC each preferred different satellite clusters (Fig 3 green

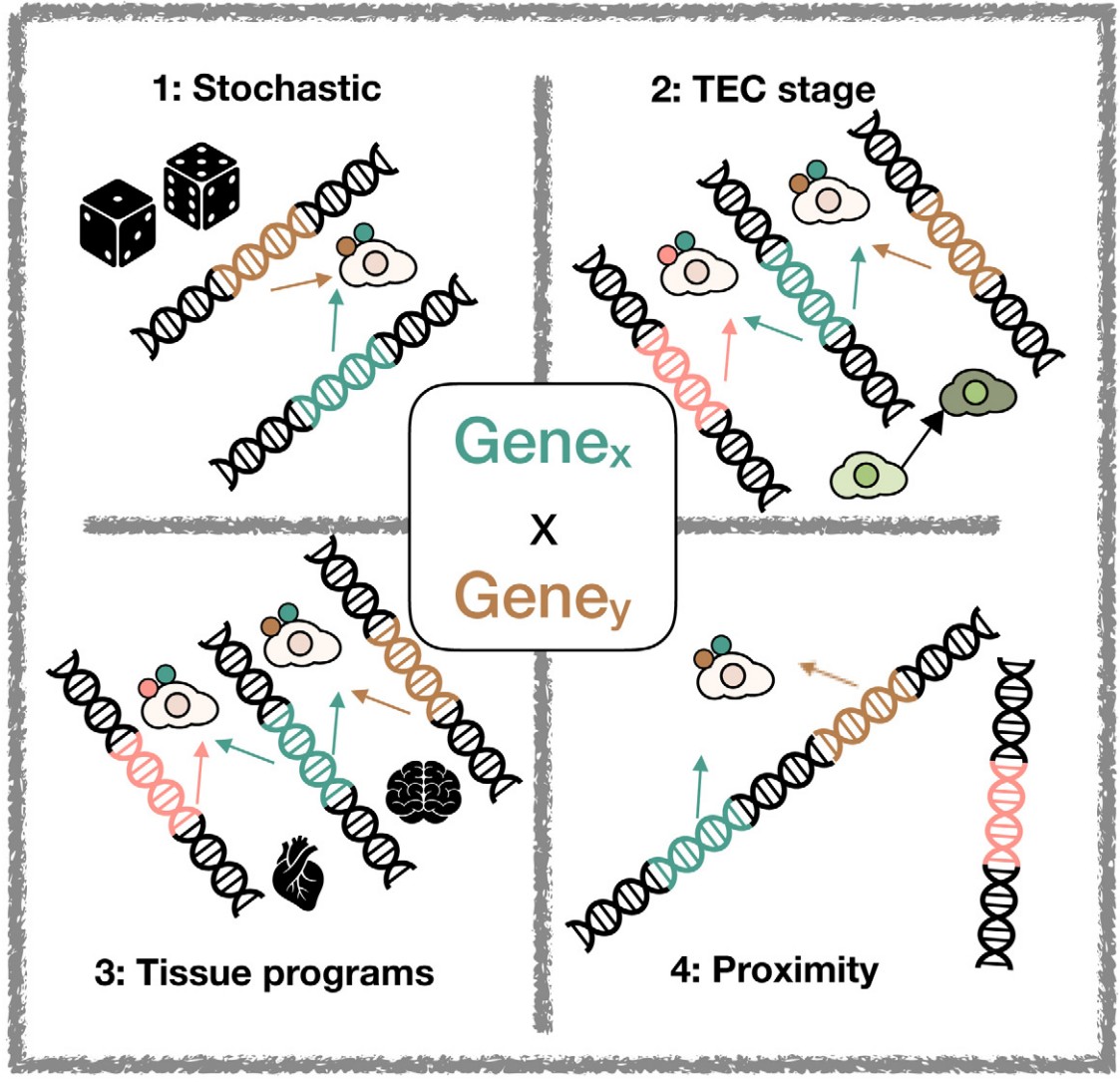

**Figure 1. Processes that could regulate TRG co-expression.**

(1) Stochastic: Gene co-expression is a fully stochastic process; (2) maturational stage: co-expression is driven by mTEC maturation stages; (3) re-use: co-expression is driven by re-use of existing tissue-restricted programmes of gene expression; (4) physical co-location: co-expressed genes are in close physical proximity.

and brown arrows in panels A, B, D). This pattern indicates that subsets of mTEC expressing a particular TRA express distinct transcriptomes non-stochastically that differentiate them from most other mTEC, a finding that is inconsistent with a Type 1 mechanism (entirely stochastic TRG expression, Fig 1).

Data were collected from three independent experiments probing different strains of mice and cell phenotypes (Table EV1). Nevertheless, each of the conditions captured the transcriptomic diversity of the batch-corrected meta-experiment (Materials and Methods; Fig 3A–D). The transition from TSPAN8$^-$/GP2$^-$ to TSPAN8$^+$/GP2$^+$ mTEC surrounded by satellite clusters is additionally evident in our analysis of published single-cell TEC datasets (Appendix Fig S3; Sansom *et al*, 2014; Brennecke *et al*, 2015; Miragaia *et al*, 2018; Bornstein *et al*, 2018). We conclude that a non-random and robustly defined set of diverse mTEC subpopulations is reproducible across multiple distinct experiments using different mouse strains and

single-cell experimental protocols and that TRG expression biases are apparent across subpopulations.

### Heterogeneity within the main mTEC body reflects their cellular maturation trajectory

Next, we considered the cells' maturational states and found that a Type 2 process (see Introduction, Fig 1) best explains PGE within single mTEC. Using unsupervised clustering, these cells resolved into 15 distinct subpopulations (Fig 4A, Appendix Fig S4A, Table EV2) which were reproducible and robust across mouse strains and unselected or selected (TSPAN8$^+$ or GP2$^+$) mTEC (Appendix Fig S4C). These subpopulations were defined by genes whose expression varies throughout mTEC maturation (Fig 4), and some of the satellite clusters were enriched for either GP2$^+$ or TSPAN8$^+$ mTEC (Appendix Fig S4A and B). Cluster definitions were

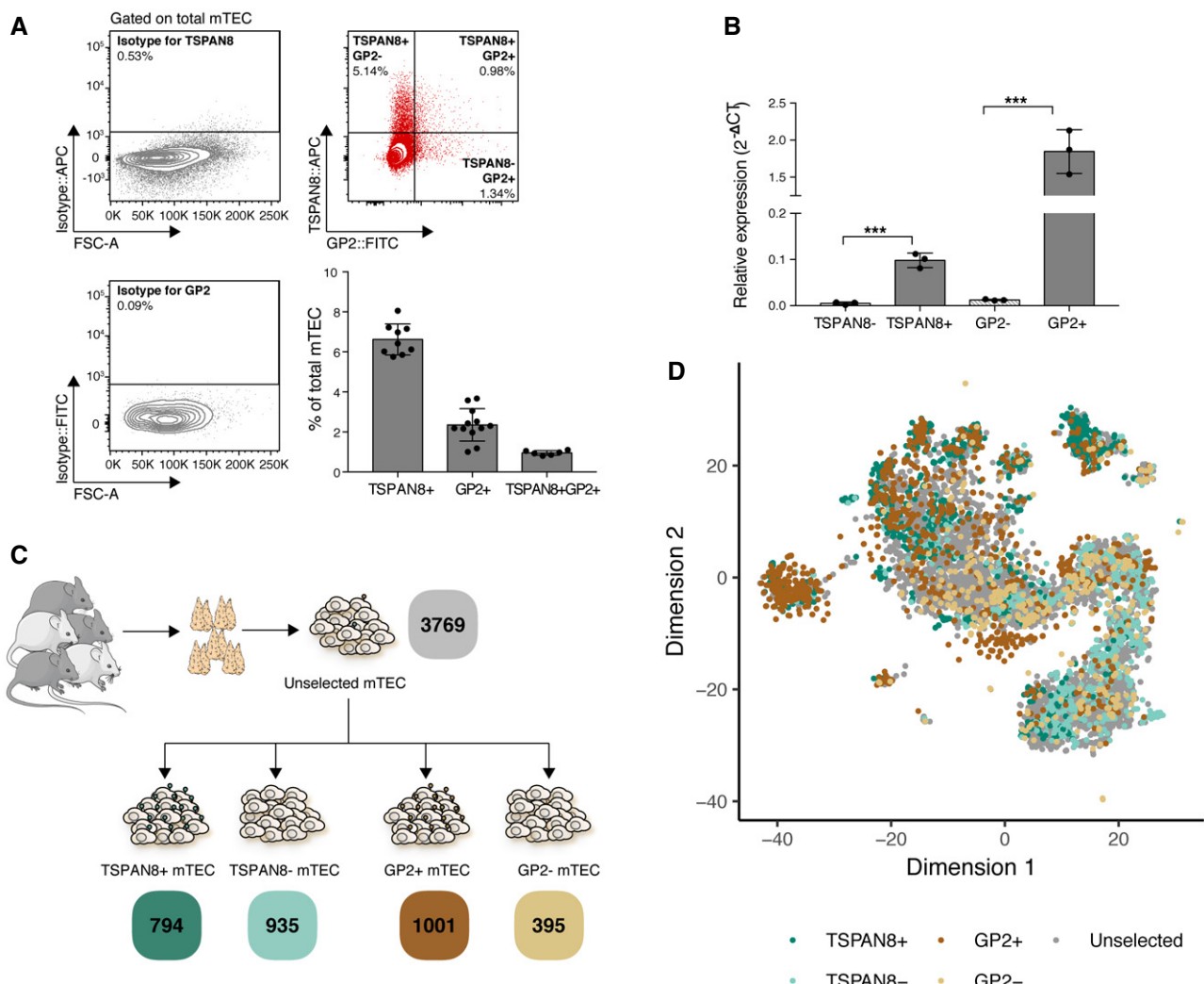

**Figure 2. Deep transcriptome analysis at single-cell resolution of thousands of flow cytometrically sorted mTEC.**

A mTEC promiscuously expressing TSPAN8 and/or GP2 (upper right panel/red) on their cell surface can be identified by flow cytometry (only final gates are shown: see Appendix Fig S1 for full gating strategy). mTEC were identified as CD45⁻EpCAM⁺Ly51⁻ (Appendix Fig S1) and the gates for TSPAN8/GP2 were set against isotype control antibodies (left panels/grey). Lower right panel: bar graph showing mean frequency (±SD) of TSPAN8⁺, GP2⁺, and TSPAN8⁺ GP2⁺ cells within total mTEC; results represent pooled data from 3 (TSPAN8⁺), 4 (GP2⁺) and 2 (TSPAN8⁺GP2⁺) independent experiments each containing three individual mice.

B Identification of TSPAN8 or GP2 protein expression via FACS reflects mRNA expression. Bar graph showing mean expression (±SD) of *Tspan8* and *Gp2* mRNA relative to *β-actin* by RT–qPCR on FACS sorted mTEC negative or positive for TSPAN8 or GP2 protein, respectively; *n* = 3, representative of two independent experiments. Significance by Students *t*-test; \*\*\**P* < 0.001.

C Schematic representation of cell populations sorted by flow cytometry for single-cell RNA-sequencing. Numbers of recovered cells are indicated below, coloured by category.

D t-SNE visualisation of mTEC subpopulations from all experiments coloured by surface phenotype established via flow cytometry; see Fig 3B and C for the distributions of unselected mTEC.

largely preserved when mTEC were clustered using all genes, or only TRGs, or when excluding TRGs (Appendix Fig S5D).

The mTEC of **clusters 1** and **2** (lower right of Fig 4A) were mainly TSPAN8⁻, GP2⁻ or unselected mTEC (Fig 3 and Appendix Fig S4A and B). For example, cluster 2 was depleted in TSPAN8⁺ or GP2⁺ mTEC based on the expected number of cells from the unselected mTEC (*P* = 0.017, 0.018 by Wilcoxon test) and enriched for TSPAN8⁻ mTEC (*P* = 0.03). Also, both clusters had little-to-no mRNA expression of *Aire* or AIRE-regulated TRGs, including *Tspan8* and *Gp2* (Fig 4C–F). Because these clusters

expressed *Pdpn* and *Ccl21a* (Appendix Fig S6), they likely represent immature junctional (Onder *et al*, 2015) and pre-AIRE mTEC (Michel *et al*, 2017). **Cluster 3** also contained mostly TSPAN8⁻, GP2⁻ or unselected mTEC (Appendix Fig S4A and B) and was similarly enriched for TSPAN8⁻ mTEC (*P* = 0.017) and depleted in TSPAN8⁺ mTEC (*P* = 0.017). Among gene markers for cluster 3 were *Mki67* and *Aire* (Fig 4G and C) whose expression contributed to the prediction (Scialdone *et al*, 2015) that these mTEC were in G₂/M-phase of the cell cycle. This cluster could, therefore, represent a proliferating subpopulation or maturational stage of mTEC.

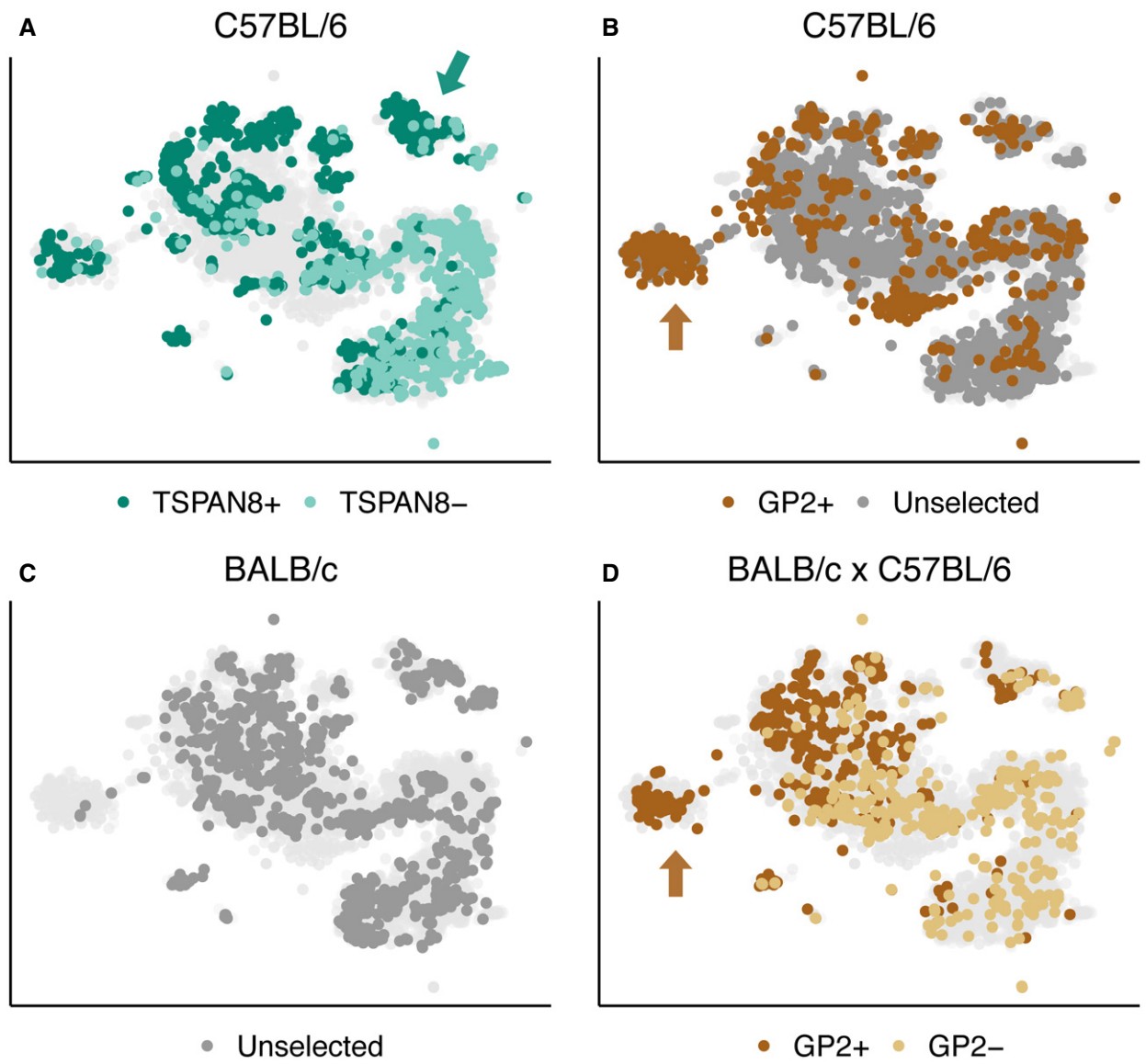

**Figure 3. Cell subpopulations are reproducible across different experiments, mouse strains and cell surface phenotypes.**

A–D   t-SNE visualisations of individual datasets from the current study, overlaid on the combined dataset (light grey). Each dot represents an individual mTEC coloured as follows: Unselected: dark grey; TSPAN8$^+$: dark green; TSPAN8$^-$: light green; GP2$^+$: dark brown; GP2$^-$: light brown. (A) 794 TSPAN8$^+$ and 935 TSPAN8$^-$ mTEC from C57BL/6 mice; green arrow identifies TSPAN8 preferred cluster. (B) 549 GP2$^+$ and 2,561 unselected mTEC from C57BL/6 mice; brown arrow identifies GP2 preferred cluster. (C) 1,208 unselected mTEC from BALB/c mice. (D) 452 GP2$^+$ and 395 GP2$^-$ mTEC from BALB/c × C57BL/6 F1 mice; brown arrow identifies GP2 preferred cluster. The whole space has good representation in most samples with the notable exception of the GP2$^+$ enriched region (brown arrow in panel B&D), which is underrepresented in the unselected cells from BALB/c due to a combination of fewer TEC analysed and the lack of enrichment for antigen-positive TEC (such as in the TSPAN8$^+$ or GP2$^+$ experiments).

**Cluster 4** represents the next likely maturational stage because these mTEC (i) were mostly negative for the expression of TSPAN8 ($P = 0.016$) or GP2 at both protein and mRNA level (Fig 4E and F, Appendix Fig S4A and B) and (ii) highly expressed *Aire* (Fig 4C) and hence also transcripts for AIRE-regulated TRGs (mean of 90 per cell). **Clusters 5** and **6** contained mTEC with the broadest TRG representation: collectively they expressed approximately 98% of detected TRGs. These mTEC not only expressed *Aire* (Fig 4C) and a high number of AIRE-regulated TRGs (mean of 82 per cell in cluster 5 and 72 per cell in cluster 6), but also *Cd80* (Fig 4B) and *Cd86*

(Appendix Fig S6), both of which function as costimulatory molecules for thymocyte activation and are expressed in mature MHCII$^{hi}$ mTEC (Michel *et al*, 2017). Moreover, they expressed TSPAN8 or GP2 protein and mRNA more frequently than clusters 1–4 (Fig 4E and F; Appendix Fig S4A and B). These features identified clusters 5 and 6 as typical representatives of PGE competent mTEC (Derbinski *et al*, 2005). The mTEC in **clusters 7** and **8** also expressed TSPAN8 and GP2 protein and their mRNA (Fig 4E and F; Appendix Fig S4A and B) as well as a moderate number of AIRE-dependent TRGs (Fig 4D). However, these two clusters had reduced or no expression

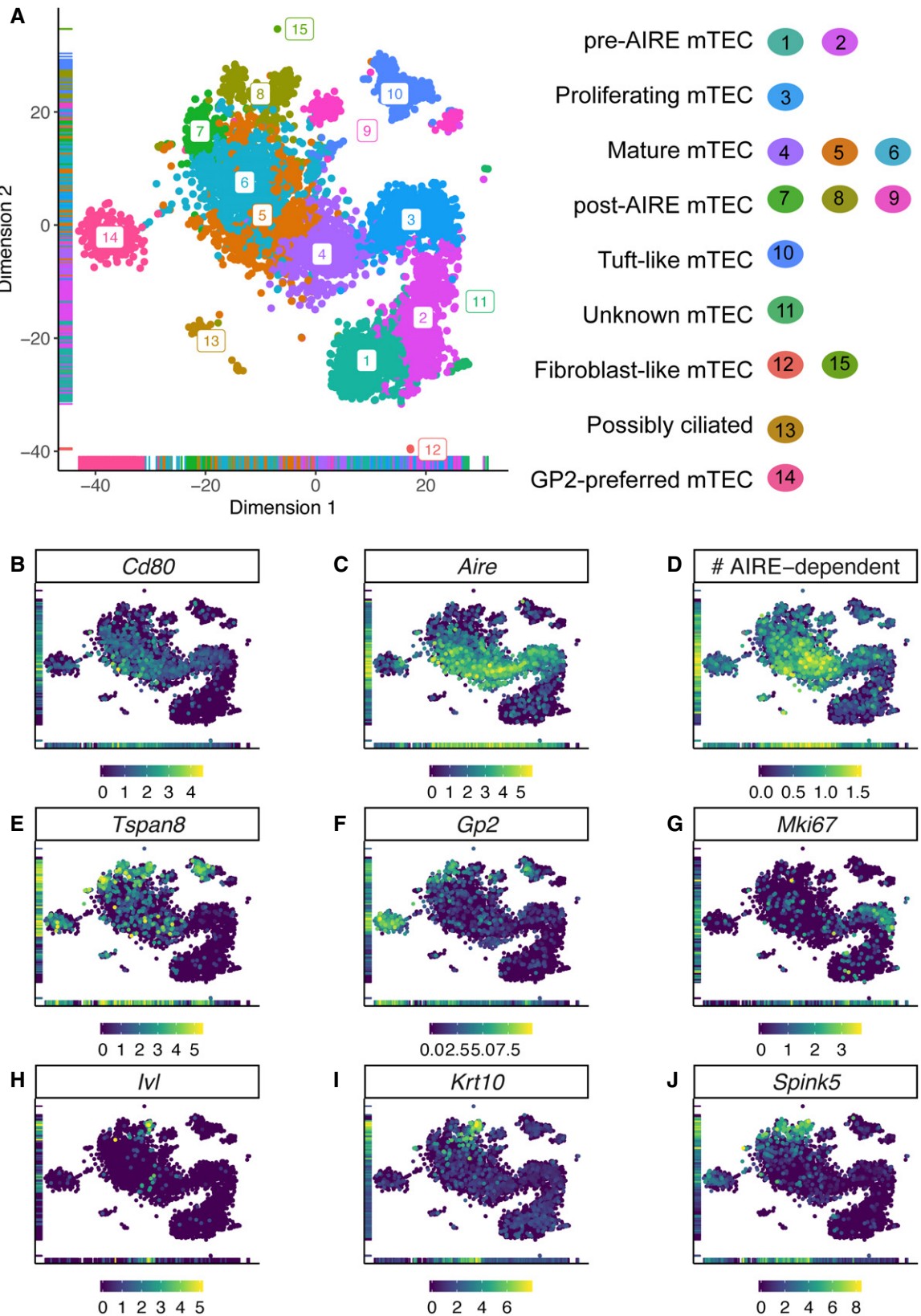

**Figure 4. Cell subpopulation clusters recapitulate known features of mTEC maturation.**

A    t-SNE visualisation of mTEC subpopulations. Each dot represents a cell coloured by cluster number.

B–J  Log2 expression level of *Cd80* (B), *Aire* (C), *Tspan8* (E), *Gp2* (F), *Mki67* (G), *Involucrin* (*Ivl*) (H), *Keratin 10* (*Krt10*) (I) and *Spink5* (J) across the dataset. The colour bar and scale beneath each plot indicate the log2 expression of the indicated gene in that cell. (D) Log10 of the number (#) of AIRE-dependent genes expressed per cell, as indicated by the colour bar and scale beneath the plot.

of *Aire*, *Cd80* and *Cd86* (Fig 4B and C, and Appendix Fig S6). In addition, they expressed markers associated with epithelial cell terminal differentiation including *Ivl* and *Krt10* (Michel *et al*, 2017; Fig 4H and I), and *Spink5* (Fig 4J). The latter has previously been found in Hassall's corpuscles (Bitoun *et al*, 2003; Galliano *et al*, 2005) and appears to be a more informative indicator of terminally differentiated mTEC in our dataset than the classically used *Ivl* and *Krt10* transcripts. In keeping with a terminally differentiated phenotype, TSPAN8 or GP2 protein positive mTEC were also significantly enriched for DSG3 expression (Appendix Fig S7), another marker associated with epithelial cell terminal differentiation found in Hassall's corpuscles (Wada *et al*, 2011; Wang *et al*, 2012).

Consistent with cells of the main body of mTEC transitioning from a TSPAN8$^-$/GP2$^-$ at its lower right pole to TSPAN8$^+$/GP2$^+$ phenotype at its upper left pole, these cells also line up along a trajectory in diffusion space (Fig 5A). We ordered the cells in pseudotime using this trajectory. In this first analysis, the two main branches in the trajectory originated at **cluster 3**, which we have inferred to be proliferating mTEC based on their expression of cell cycle relevant transcripts including *Mki67* (Fig 4G). From cluster 3, mTEC were predicted to proceed either to **cluster 4** and then **cluster 7** (Fig 5B) or **8** (Fig 5C) or, alternatively, progress via **cluster 2** to **cluster 1** (Fig 5D). An orthogonal method that uses pre- and post-spliced mRNA reads to order cells (La Manno *et al*, 2018), produced a concordant set of trajectories, with one exception, namely that the proliferating mTEC in **cluster 3** appeared to derive from **cluster 2** (Fig 5E). Taken together, these results suggest that proliferating mTEC in **cluster 3** and *Aire*$^+$ mTEC in **clusters 4–6** originated from the *Aire*$^-$*Cd80*$^-$*CD86*$^-$ mTEC in **cluster 2**. The *Aire*$^-$*Cd80*$^-$*Cd86*$^-$ mTEC from **clusters 7–8** appeared to derive from mature mTEC of **clusters 5–6** (Yano *et al*, 2008; Wang *et al*, 2012; Michel *et al*, 2017) and were transcriptionally distinct from the *Aire*$^-$*Cd80*$^-$*Cd86*$^-$ cells in **clusters 1** and **2**. Consequently, we propose that **clusters 1** and **2** represent pre-AIRE mTEC (distinguished by *Ccl21a* and *Pdpn* expression), while **clusters 7** and **8** represent likely post-AIRE mTEC (distinguished by *Ivl, Krt10* and *Spink5* expression). These findings are in keeping with current models of mTEC maturation (Sun *et al*, 2013; Michel *et al*, 2017).

Two of our main observations are, to our knowledge, novel. Firstly, pre-AIRE mTEC can be distinguished into two subtypes (clusters 1 and 2), and secondly, the mTEC in **cluster 2** precede those in **cluster 1** in the pseudotime analysis. One potential interpretation of this is that the **cluster 1** mTEC represent a class of quiescent progenitor TEC (Wong *et al*, 2014), while **cluster 2** mTEC represent MHCII$^{lo}$ mTEC that are transitioning to MHCII$^{hi}$ mTEC. The mTEC of **cluster 1** highly express genes such as *Itga6* (CD49f) and *Ly6a* (Sca-1) that are markers of a quiescent mTEC progenitor population with limited regeneration potential (Wong *et al*, 2014).

### Aire$^-$/low satellite clusters lie separately from the main trajectory of mTEC maturation

Seven satellite clusters surround the main body of mTEC as displayed by the t-SNE visualisation (Fig 4A). **Cluster 9** (split into 2 sub-clusters) appeared to contain mTEC involved in negative regulation of proliferation (Gene Ontology analysis) and in pathways dealing with the response to stress (Reactome analysis; Appendix Fig S8). These mTEC were a mixture of TSPAN8 and GP2 positive and

negative cells and were largely devoid of *Aire* expression. **Cluster 10** contained TEC recently labelled as "thymic tuft cells" (Bornstein *et al*, 2018; Miller *et al*, 2018). This cluster was significantly enriched for TSPAN8$^+$ mTEC (Fig 3A, and Appendix Fig S4A and B; $P = 0.017$) and was characterised by the expression of *Pou2f3*, a transcription factor involved in regulating tuft cell function in the intestinal epithelium and respiratory tract (Appendix Fig S6; Reid *et al*, 2005; Yamashita *et al*, 2017) and of its target genes including *Tas2r* genes and *Trpm5* (Yamashita *et al*, 2017). **Cluster 11** was characterised by the expression of genes related to RNA metabolism and nonsense-mediated decay (Appendix Fig S8) and **cluster 12** by the expression of genes related to the organisation of the extracellular matrix (Appendix Fig S8). Both clusters 11 and 12 were mainly TSPAN8$^-$ and GP2$^-$ mTEC. **Cluster 13** cells express genes involved in the response to stress and external stimuli as well as cilium assembly (Appendix Fig S8) and are mostly TSPAN8$^+$ or GP2$^+$ mTEC. **Cluster 14** was enriched for GP2$^+$ mTEC (Appendix Fig S3B; $P = 0.010$) and expressed low levels of *Aire*. Other markers of this cluster included the chemokine ligands *Ccl6* (Appendix Fig S6), *Ccl9* and *Ccl20* and chemokine receptor type 5 (*Ccr5*) suggesting a potential role in cell communication. Finally, **Cluster 15** was characterised by the expression of genes related to the organisation of the extracellular matrix (Appendix Fig S8).

Using FACS to enrich for TSPAN8$^+$ mTEC and GP2$^+$ mTEC, respectively, enabled us to investigate a large number of rare **cluster 10** and **14** cells. Nearly half the mTEC in these clusters were positive for their respective TRAs (44 and 49%, respectively) and the next largest contributor to these clusters was unselected cells for which we have no measurement of TSPAN8 or GP2 protein levels (37 and 39%, respectively). Importantly, these clusters were robust to clustering unselected mTEC on their own (Fig 3C). Furthermore, while **cluster 10** contained thymic tuft cells (Bornstein *et al*, 2018; Miller *et al*, 2018), **cluster 14** was transcriptionally distinct and expressed a set of chemokine ligands and receptors that are absent from **cluster 10**.

These observations argue for an uneven expression of TRGs across mTEC subpopulations, implying that satellite clusters show preference for expression of particular gene subsets and providing additional evidence against a Type 1 process (TRG expression is entirely stochastic, Fig 1), and in favour of a Type 2 process (different maturational stages or classes of mTEC activate TRG expression differentially, Fig 1).

### Gene module clustering reveals robustly identified gene co-expression groups supportive of PGE being an ordered process

The robust identification of 15 distinct mTEC clusters, which were largely preserved when clustering was performed using only TRGs (Appendix Fig S5D), suggested an ordered process that selects TRGs to be co-expressed within single mTEC. To further investigate the patterns of gene co-expression, we applied a method of gene module clustering that accentuates expression similarities found among rarely observed genes and attenuates similarities found among frequently observed genes. This allowed us to assign genes to single gene modules that exhibited a distinctive gene co-expression profile across our dataset. In total, 14,861 genes were assigned to 50 modules (Fig 6A). A further 7,958 genes, including many olfactory and vomeronasal receptor genes, could not be assigned to a module

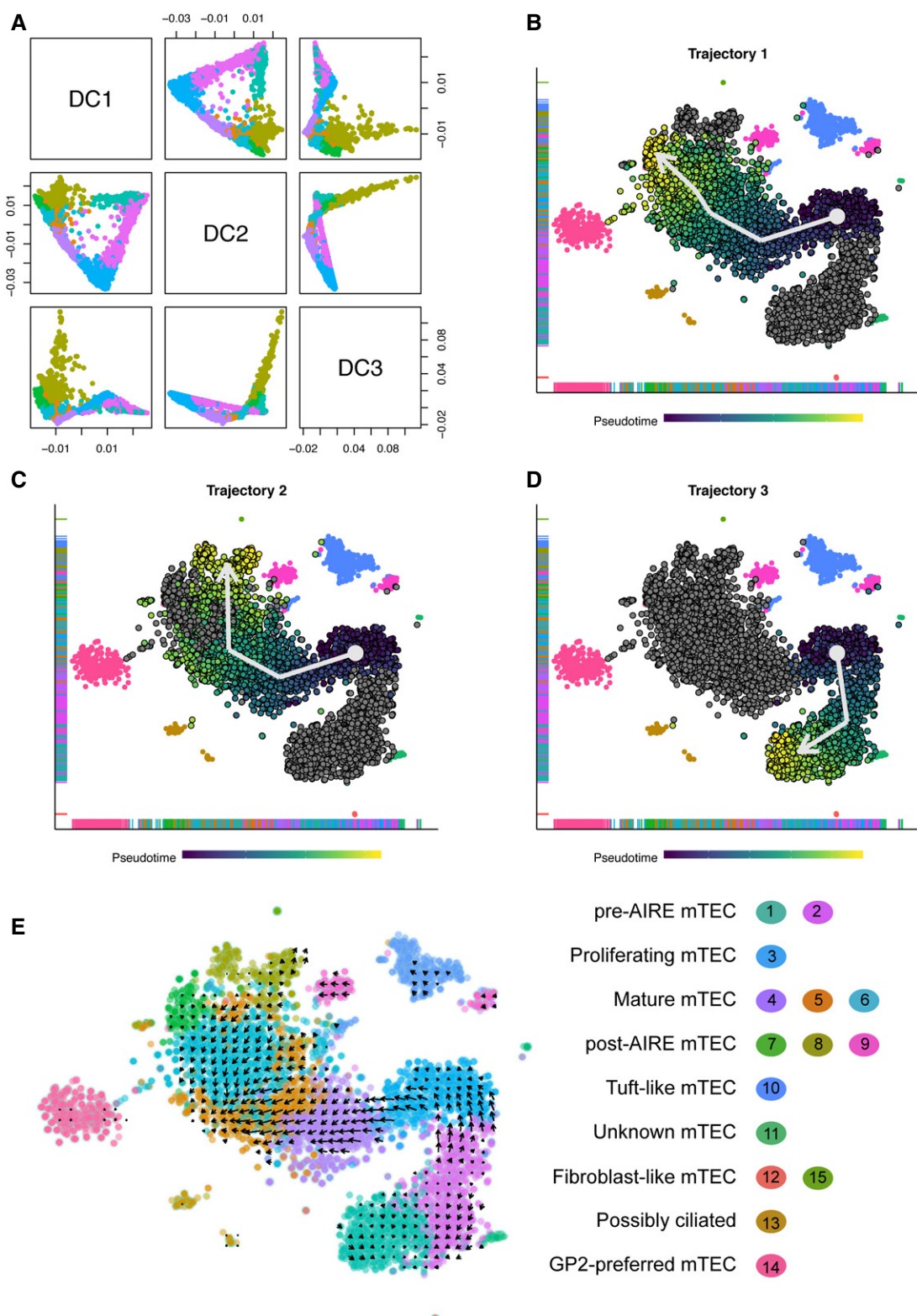

**Figure 5. Pseudotemporal ordering of mTEC resolves three trajectories of mTEC maturation.**

A Paris plot of the first three diffusion components (DC) of the dataset described in Fig 4A.

B–D t-SNE visualisation of mTEC subpopulations. Each dot represents a cell coloured by the inferred position in the pseudotemporal ordering. A white arrow indicates the direction of each trajectory. In each plot, the cells along the trajectory are coloured by their position along the inferred pseudotemporal trajectory with dark blue and yellow indicating the extreme ends of the trajectory in question.

E t-SNE visualisation of mTEC subpopulations with arrows showing the ordering as identified by RNA velocity analysis. Each dot represents a cell coloured by cluster ID (right).

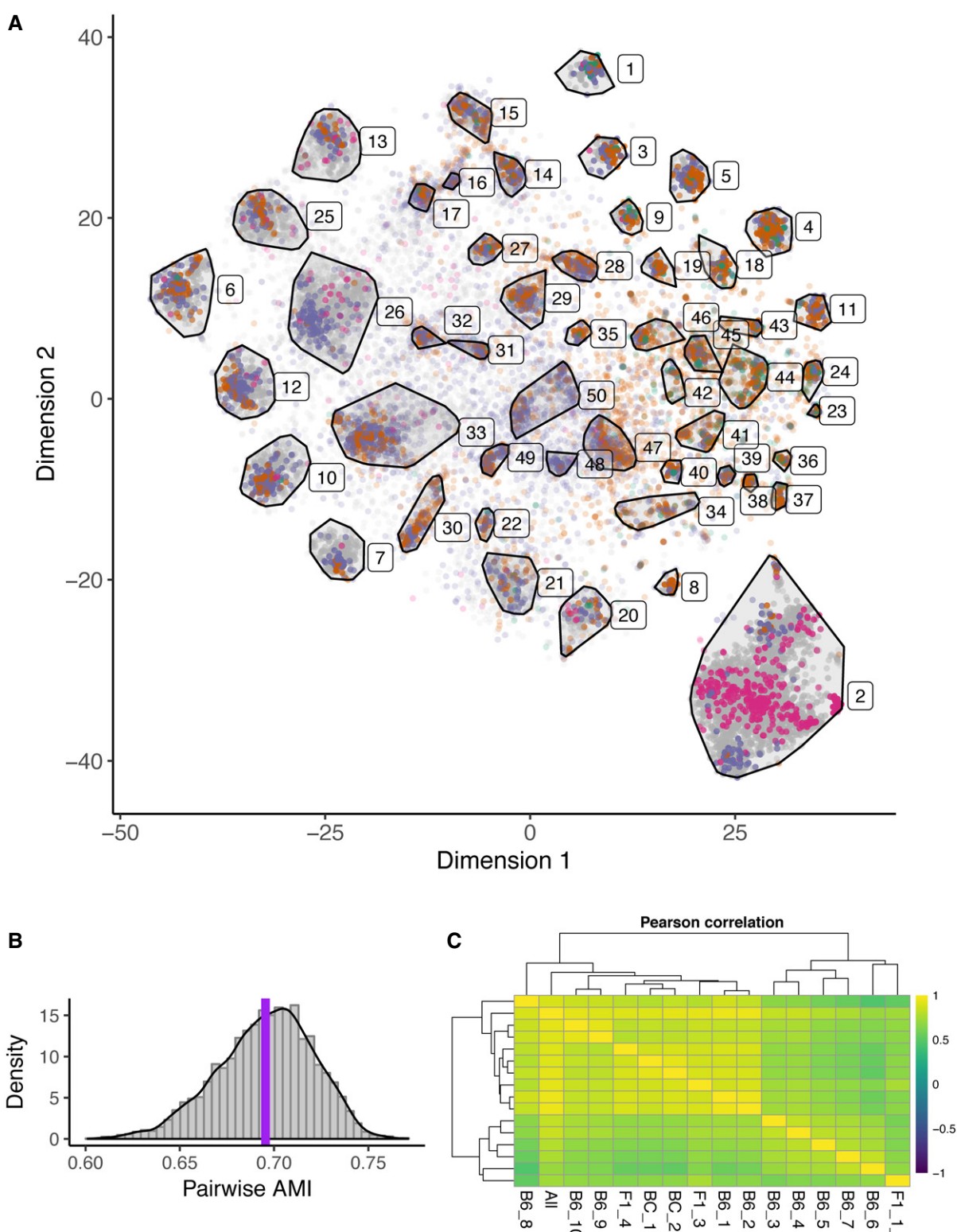

**Figure 6. Robust gene co-expression modules indicate that mTEC gene co-expression has order.**

A  t-SNE visualisation of the partition of 22,819 genes into 50 mutually exclusive gene modules. Each dot represents a gene and is coloured according to gene expression category (AIRE-dependent, AIRE-enhanced, AIRE-independent TRG, Other, Housekeeping, Unclassified) as described in Fig 7A. The colour intensity of each dot/gene is proportional to its membership probability within each module.

B  Histogram of adjusted mutual information (AMI) between random samples. The mean AMI value is indicated by a vertical purple line; AMI values lie between 0 and 1.

C  Pearson correlation of co-expression frequency for pairs of TRGs (AIRE-regulated and AIRE-independent) between individual mice or all mice pooled together (see Table EV1 for mouse identifiers). Mean correlation = 0.75, with the largest divergence observed between samples with higher read depth (B6_3-B6_7, and F1_1_2).

due to their low detection frequency. The definition of these 50 gene modules was both reproducible and robust because re-clustering of 100 random subsamples of the data produced highly similar modules (mean adjusted mutual information (AMI) score = 0.695; Fig 6B; Table EV3). This finding is not explained by cellular expression levels because the analysis used a TF-IDF transform (Manning *et al*, 2008) ensuring that frequently expressed genes do not contribute substantially to the module identities.

Next, we sought to determine the variability of TRG co-expression between individual mice (of either the same or different genetic backgrounds). For each pair of TRGs, the fraction of mTEC expressing both TRGs was calculated per mouse. These fractions were then compared across all mice in our dataset. The mean Pearson correlation of these fractions across all mouse pairs was high (average value of 0.75; Fig 6C). To further investigate whether the TRG co-expression frequencies were driven by the likely post-AIRE mTEC, we recomputed these frequencies as above using only mTEC from clusters 3–6 (likely AIRE$^+$ mature mTEC; Appendix Fig S5C). This restricted analysis showed that TRG co-expression frequencies remained highly correlated (mean Pearson correlation of 0.69). Our findings thus demonstrate that sets of TRGs were repeatedly co-expressed in individual mTEC and that this TRG co-expression was replicated both in random subsets of mTEC and across different mice.

Half of the 50 gene modules were significantly enriched in TRGs (53% median proportion of AIRE-regulated or AIRE-independent TRGs across these modules; Fig 7A–C). By contrast, seven gene modules had significantly fewer TRGs than expected by chance (15% median TRG proportion). Genes in each module were expressed at a comparable level (normalised UMI count) and within similar mTEC subsets across the 6,894 single cells analysed. These observations imply that complex gene co-expression programmes were replicated in individual modules across many mTEC and were driven by both TRGs and by non-TRGs. The significant contribution of TRGs to half of the gene modules further implied that their co-expression substantially defined these modules. For example, *Gp2* was assigned to module 7 and genes in module 7 were most highly expressed in the likely post-AIRE clusters 7 and 8, as well as in the GP2-preferred cluster 14. In contrast, *Tspan8* was assigned to module 31, whose member genes were most highly expressed in the tuft cell-like cluster 10. Consequently, although both TSPAN8$^+$ and GP2$^+$ mTEC were located in some of the same cell clusters, *Tspan8* and *Gp2* were co-expressed with very different sets of TRGs. This further supports the theory that PGE results in ordered gene co-expression.

Three modules contained genes frequently expressed in mTEC that were not significantly enriched for TRGs (Fig 7B and C). One of these modules (module 2) contained genes that were detected, on average, in approximately 45% of mTEC, whereas genes in modules 26 and 48 were expressed in about 12% of mTEC and all three contained few AIRE-regulated TRGs. Several modules contained genes highly expressed in a particular mTEC maturational state or subpopulation. For example, module 26 contained genes expressed in the cycling cells of cluster 3 (including *Mki67*, cyclins and *E2f* genes) and module 48 encompassed genes expressed in the *Aire*-expressing clusters 4 and 5. Modules 31 and 32 included genes that are characteristic of thymic tuft cells (cluster 10) such as the *Tas2r* family, *Il10*, *Il25* and *Dclk1*, and genes co-expressed under the transcriptional control of POU2F3 (Yamashita *et al*, 2017; Appendix Fig S6). Module 49 contained transcripts of chemokines (including *Ccl6*,

*Ccl9* and *Ccl20*) and chemokine receptors (*Ccr1, Ccr2* and *Ccr5*) typical of cluster 14 (Appendix Fig S6).

In summary, 50 gene co-expression modules were identified, half of which were largely driven by TRG co-expression patterns reproducible in different mice. This again argues against an entirely stochastic mechanism for TRG expression within single mTEC (Fig 1).

### While gene expression in mTEC has order, gene membership in co-expression clusters is biologically indeterminate

Next, we asked what feature, such as chromosomal location, intergenic distance, tissue-, pathology- or pathway-restricted expression, might explain the observed order of TRG co-expression in single mTEC. The co-expression pattern identified in the majority of the gene modules was nearly always independent of expression by a single chromosome (Fig 7D), with the exception of modules 33 and 44, which contained more transcripts of genes located on chromosome 3, and modules 32 and 35, which had a higher frequency of transcripts from genes on chromosomes 6 and 7 than expected. Nevertheless, such enrichments are not highly explanatory of the co-expression order observed in 46 of the 50 gene modules.

The chromosomal distances between genes present within the same gene module were near identical to those of all other genes regardless of AIRE dependency (Fig 7E). Moreover, we found no evidence that individual modules were biased in their gene expression for (i) TRAs of individual peripheral tissues (Fig 7F), (ii) antigens characteristic of individual organ-specific autoimmune pathologies (Fig 7G) or (iii) molecules assigned to a particular cellular pathway or Gene Ontology term. These findings are not consistent with a Type 3 mechanism in which mTEC recapitulate the transcriptional programme of a peripheral tissue (Fig 1).

Finally, we considered whether the transcriptomic identity of mTEC varied according to their spatial location. For this, we computed the G-function, the cumulative distribution function of distances between nearest neighbour mTEC each expressing the same TRA, here either TSPAN8 or GP2 (Fig 8A and B, and Appendix Fig S9). For comparison, G-functions were also computed for simulated mTEC that are (i) evenly spaced, (ii) dispersed randomly or (iii) clustered (Fig 8B and C). Our results showed that either TSPAN8$^+$ or GP2$^+$ mTEC are spatially distributed within the medulla in a manner consistent with a random process (Fig 8B and Appendix Fig S9).

Together, these results indicate that PGE in the medulla is a biologically indeterminate yet ordered process whose order is provided by repeated co-expression of particular gene subsets. Furthermore, these co-expressed genes are not systematically collocated in the linear genome and are not linked by tissue specificity, or by biological or disease processes.

## Discussion

The molecular processes underlying PGE in single mTEC have remained unclear despite numerous studies addressing this topic. These studies, which have been limited by technology and cell number, have not established conclusively whether TRG expression in single mTEC is stochastic or ordered (Derbinski *et al*, 2005, 2008; Villaseñor *et al*, 2008; Pinto *et al*, 2013; Sansom *et al*, 2014; Brennecke *et al*, 2015; Meredith *et al*, 2015; Rattay *et al*, 2016).

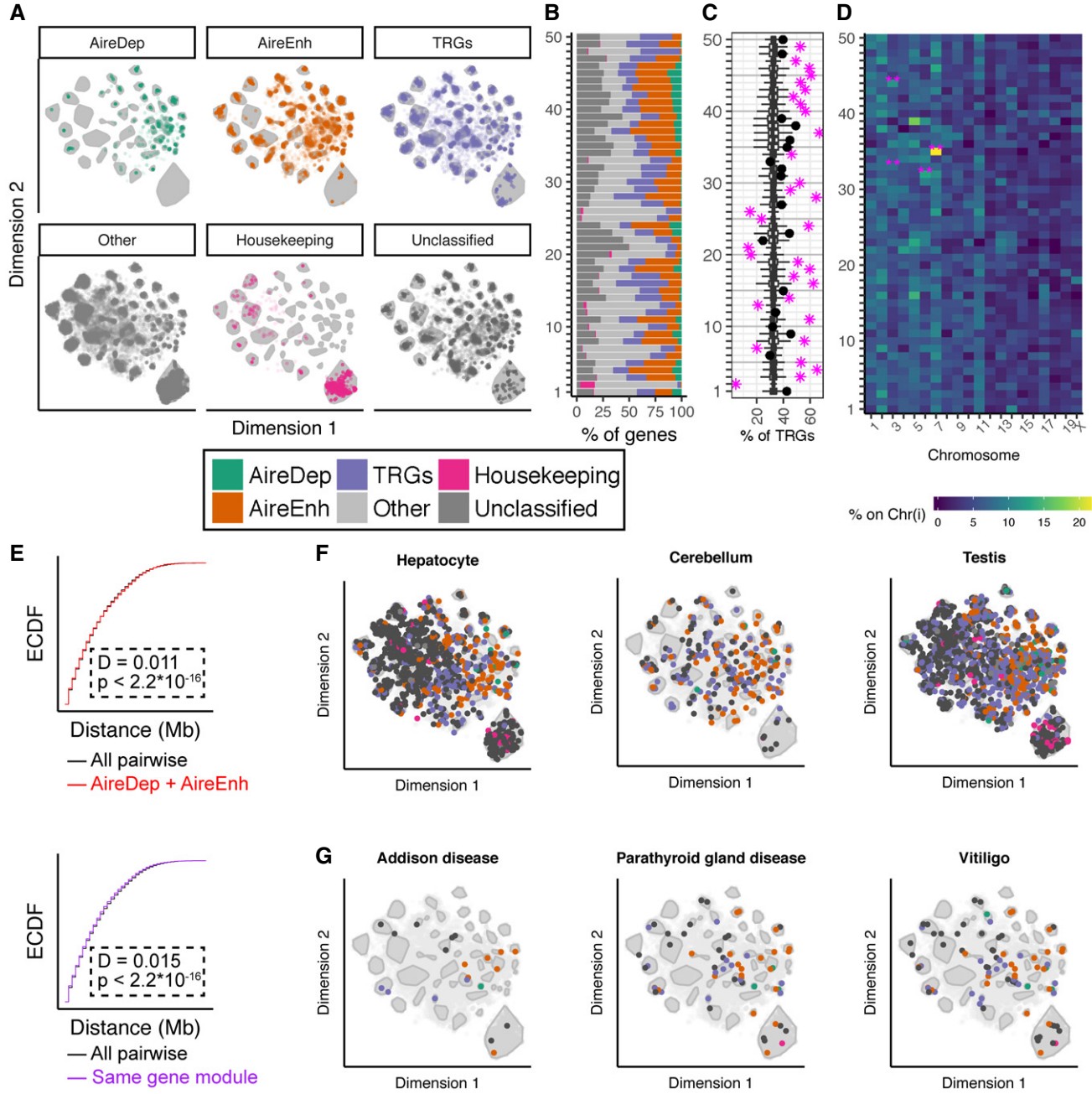

**Figure 7. Randomness underlies the structure in gene co-expression modules.**

A  Visualisation of the composition of gene modules coloured by gene expression category (AIRE-dependent, AIRE-enhanced, AIRE-independent TRG, Other, Housekeeping, Unclassified).

B  Gene expression category composition of each gene module displayed as a stacked bar chart.

C  Percentage of TRGs in each module. Modules significantly enriched or depleted in TRGs are indicated by a magenta star and non-significant modules by a black dot. Boxplots show the expected contribution of TRGs from 10,000 random samples. Each boxplot is depicted with the median as a central bar, the extent of the box as the 25[th] and 75[th] percentiles and the whiskers extending from the box to either the farthest value or no more that 1.5 * the interquartile range.

D  Chromosomal locations for genes expressed within each gene module. **Adjusted $P$-value < 0.01 (in pink) indicates that more genes assigned to a given module ($x$-axis) are located on a particular chromosome ($y$-axis) than expected by chance (empirical $P$-value, corrected for multiple comparisons).

E  Empirical cumulative distribution function (ECDF) for distribution of the pairwise genomic distance between co-expressed genes. Top: all pairwise distances in grey; between AIRE-regulated genes only in red; bottom: all pairwise distances in grey; between genes within the same gene module only in purple.

F  Distribution of tissue-specific genes across gene modules for hepatocytes (left), cerebellum (middle) and testis (right); tissue-specific genes are highlighted with a coloured dot that denotes the gene category. Kolmogorov–Smirnov test used to compare ECDFs (E, F).

G  Distribution of disease-related auto-antigen genes across gene modules for parathyroid gland disease (left), Addison's disease (middle) and vitiligo (right); auto-antigen genes are highlighted with a coloured dot that denotes the gene category.

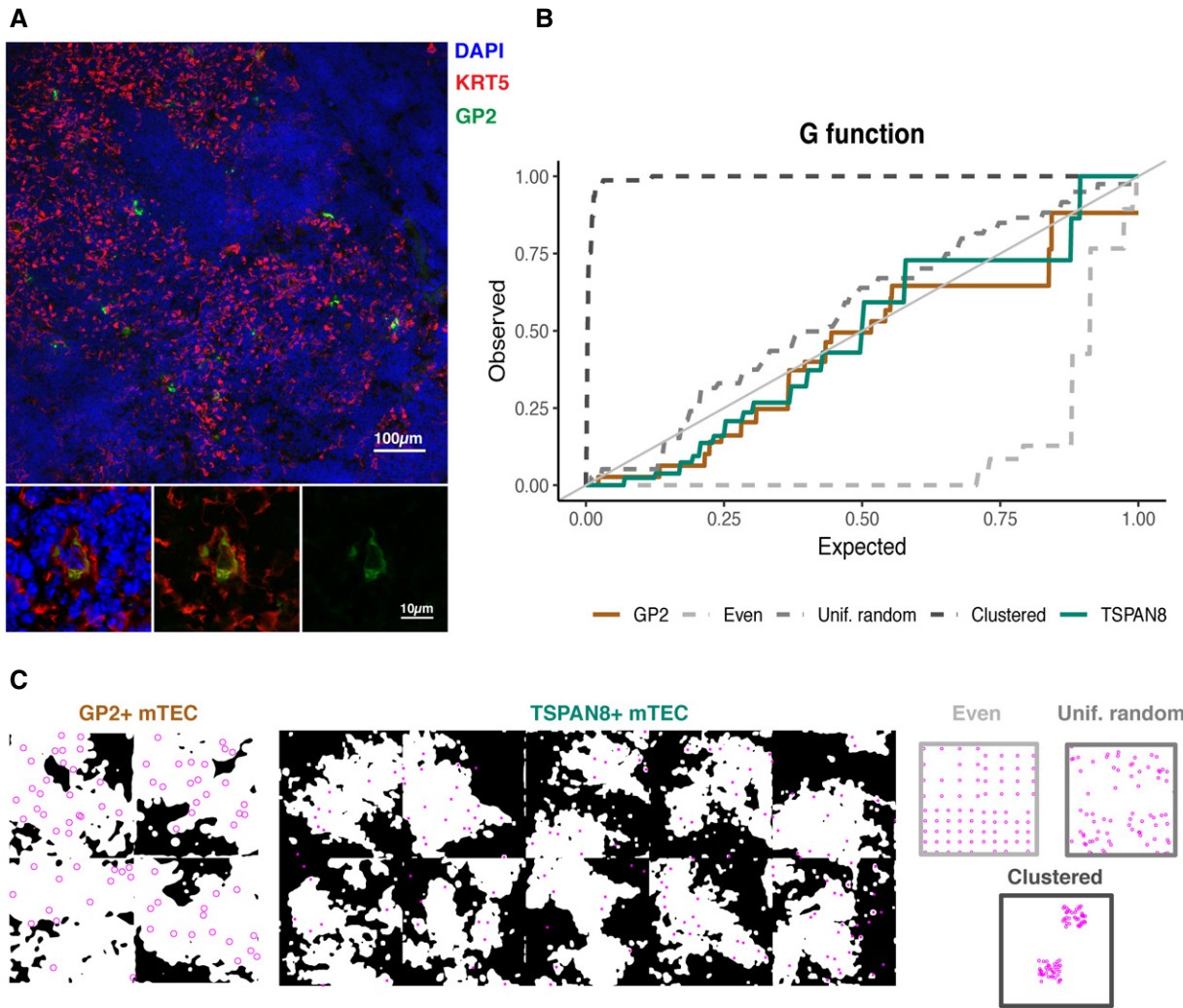

**Figure 8. GP2+ or TSPAN8+ mTEC are not spatially clustered but instead dispersed randomly within medullary islands.**

A Representative microscope image showing GP2+ mTEC. GP2 is stained in green and KRT5, which marks mTEC, in red. Scale bar 100 µm. Zoomed images in the lower panels show the overlap of GP2 and KRT5 for one GP2+ mTEC. Scale bar 10 µm.

B Observed vs. expected G-function for GP2 (brown) and TSPAN8 spacing (green) and exemplar spacings.

C Masked medullary region (white) for four slides stained as in (A) with GP2+ mTEC as magenta circles and ten slides stained for TSPAN8, with TSPAN8+ mTEC as magenta dots (note: the upper left GP2+ mask corresponds to the image in panel A). Exemplar distributions for comparison (coloured as in B).

Some studies that failed to detect co-expression patterns used single-cell PCR limited to the detection of only a very small number of TRGs in a small number of cells (Derbinski *et al*, 2008; Villaseñor *et al*, 2008). Co-expression patterns were detectable in three studies that either preselected mTEC for the expression of cell surface-expressed TRAs (Pinto *et al*, 2013) or conducted single-cell RNA-sequencing of hundreds of single mTEC (Brennecke *et al*, 2015; Meredith *et al*, 2015). Two studies found that mTEC express a larger repertoire of TRGs as they mature and that single mTEC are not restricted to expressing TRGs belonging to particular peripheral tissues, biological pathways or controlled by the same transcription factors (Pinto *et al*, 2013; Meredith *et al*, 2015).

Our analysis of thousands of single mTEC revealed differentiating gene co-expression patterns that are maintained not only among individual mice of the same genetic background but also in different mouse strains and that are also observed under different experimental designs (selected vs. unselected) and batches. In aggregate, these results strongly suggest that an ordered process regulates PGE in mTEC.

Cell-based clustering of our data supports and expands significantly upon previous observations that the mTEC compartment is heterogeneous (Bornstein *et al*, 2018; Miragaia *et al*, 2018), a notion supported by studies tracking TEC differentiation during organogenesis and regeneration (Gäbler *et al*, 2007; Yano *et al*, 2008; Wang *et al*, 2012; Metzger *et al*, 2013; Ohigashi *et al*, 2013, 2015; Nishikawa *et al*, 2014; Mayer *et al*, 2016). These suggest that mTEC derive during embryogenesis from progenitors expressing claudin-3 and claudin-4 and SSEA-1 (Hamazaki *et al*, 2007; Sekai *et al*, 2014), and postnatally from a population localised at the corticomedullary junction expressing podoplanin and CCL21 (Onder *et al*, 2015;

Mayer *et al*, 2016; Michel *et al*, 2017). Immature mTEC that are AIRE⁻CD80⁻CD86⁻MHCII^lo transition to an AIRE⁺CD80⁺CD86⁺ MHCII^hi state that represents the most functionally mature mTEC from the perspective of PGE (Gäbler *et al*, 2007). Finally, mTEC lose expression of AIRE, MHCII and costimulatory molecules and enter a terminally differentiated state (Yano *et al*, 2008; Wang *et al*, 2012; Metzger *et al*, 2013; Nishikawa *et al*, 2014).

In keeping with the above model of mTEC maturation, two clusters (clusters 1 and 2) were identified that display progenitor-like characteristics because they express *Ccl21a* and *Pdpn* but lack *Aire*, *Cd80* and *Cd86* and are positive for transcripts encoding p63 (*Trp63*; Appendix Fig S6), a TEC lineage-specific determinant of proliferative capacity, *Itga6* (Cd49f; Appendix Fig S6), an integrin α-chain essential for TEC adhesion (Golbert *et al*, 2013) and *Sca-1* (Ly6a; Appendix Fig S6), a marker identifying a TEC progenitor status (Wong *et al*, 2014; Ulyanchenko *et al*, 2016). Custer 1 also expresses eotaxin-1 (*Ccl11*; Appendix Fig S6), a chemokine known to attract eosinophils via the chemokine receptor Ccr3 suggesting a possible interaction between these mTEC and thymic resident eosinophils (Garcia-Zepeda *et al*, 1996; Matthews *et al*, 1998; Throsby *et al*, 2000; Kim *et al*, 2010). Pre-AIRE immature mTEC (cluster 2) transit to an *Aire*⁺*Cd80*⁺*Cd86*⁺ (clusters 5 and 6) phenotype through an actively cycling stage (cluster 3), and then to a likely post-AIRE stage (clusters 7 and 8) with low *Aire, Cd80,* and *Cd86* and high *Krt10, Ivl* and *Spink5* expression.

Single-cell analyses recently defined distinct mTEC subpopulations. Bornstein *et al* (2018) identified four classes of mTEC (labelled mTEC I–IV) that largely agree with our data (Appendix Fig S3F). Clusters 1 and 2 relate to the immature pre-AIRE mTEC I subpopulation, clusters 3 and 6 to the mature AIRE⁺ mTEC II, clusters 7 and 8 to mTEC III, and cluster 10 to the tuft-like mTEC IV (Appendix Fig S3F; Bornstein *et al*, 2018).

We also identified six novel mTEC clusters that warrant further experimental characterisation in the context of future studies. Cluster 14 constitutes the largest newly described subpopulation (436 observed mTEC, 4.6% of unselected mTEC), enriched for mTEC expressing GP2 and defined by high expression of chemokine ligands *Ccl6* (Appendix Fig S6), *Ccl9* and *Ccl20*, as well as the chemokine receptor *Ccr5*. This result suggests that our selection of rare mTEC subtypes, using FACS enrichment, has revealed otherwise hidden subpopulations and thus that analyses of mTEC expressing other TRAs would likely uncover additional satellite clusters, each with a distinct transcriptome. Our second largest novel cluster is cluster 9 (227 cells), which expressed the markers *Ceacam10, Cd177* and *Ckm*. Cluster 9 shares transcriptome similarities not only with the terminally differentiated mTEC of cluster 8 but also with tuft-like mTEC of cluster 10. A third novel cluster, cluster 13 contains 117 mTEC and highly expresses several genes associated with cilium assembly (*Spag16*, *Wdr34* and *Bbs7*). The remaining novel clusters (clusters 11, 12 and 15) are defined by few cells (0.3–1% of the collected mTEC) and remain largely uncharacterised. The cells in these clusters retained a strong signature of expression of core mTEC genes (gene module 2), and while we cannot rule out contamination by other classes of TEC (cortical TEC) or technical artefacts such as doublets, our repeated observation of these cells across multiple experiments combined with their similarity to other TEC suggested that they represented a rare subpopulation of TEC rather than a contaminant.

Low cell number has been a limitation of previous single-cell studies and is expected to have limited the detection of gene co-expression groups because of the low frequencies at which individual TRGs are expressed in single mTEC (Sansom *et al*, 2014; Brennecke *et al*, 2015; Meredith *et al*, 2015). In contrast to previous studies, which were either unable to detect co-expression (Derbinski *et al*, 2008; Villaseñor *et al*, 2008; Sansom *et al*, 2014), or were able only to detect a handful of co-expression groups (Pinto *et al*, 2013; Brennecke *et al*, 2015; Meredith *et al*, 2015), we were able to detect 50 separate gene co-expression modules, the majority of which comprised TRG co-expression sets (48 out of 50; Figs 6A, and 7A and B), suggesting that PGE in single mTEC follows an ordered process. Nevertheless, approximately one-third of the genes expressed in multiple cells could not be assigned to a co-expression group. Although this could reflect biological noise, it is possible that, with the inclusion of more single cells, these genes might also become assignable to specific gene modules.

A previous study found that co-expression patterns differed between individual mice and suggested that gene co-expression networks were established stochastically in each mouse (Meredith *et al*, 2015). This study investigated only 200 mTEC isolated from two pairs of WT or *Aire* knockout mice, raising a concern that it was significantly underpowered. By contrast, the gene modules identified in our study were robust as evidenced by their reproducibility across random samples of the dataset (Fig 6B), as well as between individual mice independent of their strain (Fig 6C). Such reproducible TRG co-expression is likely to reflect an inherent cellular property that introduces bias in which TRGs are expressed together within a cell. That being said, we were unable to explain membership to co-expression groups by any of the features we examined, including gene category, chromosome localisation, genomic distance, tissue specificity, autoimmune disease association and inclusion in a specific biological pathway. The largely random distribution of genes in modules across the genome is in keeping with observations from previous studies (Pinto *et al*, 2013; Meredith *et al*, 2015; Miragaia *et al*, 2018). Our results are also consistent with previously published studies that concluded that PGE patterns in mTEC are dictated neither by expression patterns seen in differentiated peripheral cell types nor by co-regulation by specific transcription factors (Villaseñor *et al*, 2008; Pinto *et al*, 2013; Meredith *et al*, 2015). While we were unable to explain membership to co-expression groups, this does not preclude the existence of an underlying mechanism that explains why certain TRGs are co-expressed in single mTEC particularly since inter-individual reproducibility was robust.

In the introduction, we presented four molecular processes that could account for the heterogeneity of PGE in single mTEC. Our results provided evidence against TRG expression being entirely stochastic (Type 1, Fig 1) and also show that co-expression patterns in mTEC are not driven by the same cellular processes as in peripheral tissues (Type 3, Fig 1) or by contiguous co-location of co-expressed genes on the same chromosome (Type 4a, Fig 1). By contrast, our data provided evidence that different maturational stages or classes of mTEC activate TRG expression differentially (Type 2, Fig 1) and our data do not exclude the possibility that TRGs are physically co-located on chromatin by virtue of chromatin looping (Type 4b, Fig 1), as has been suggested by previous studies. Specifically, Pinto *et al* (2013) demonstrated that co-expressed TRGs are co-localised within the same nuclear subdomains via DNA-FISH

and Bansal *et al* (2017) showed that AIRE acts at superenhancers, which are known to localise to the genes they regulate via looping.

The biologically indeterminate yet ordered process for TRG co-expression described here suggests a system in which antigens are presented in a randomly dispersed manner across the medulla under a programme that repeatedly generates mTEC with co-expressed genes that are randomly sampled with respect to disease-relevant antigens, pathways, tissues and chromosomes. Furthermore, the spatial locations of TSPAN8$^+$ and GP2$^+$ mTEC are randomly dispersed across the thymic medulla. Should the observed random spatial distribution of TSPAN8 and GP2 presentation be generalisable to other TRAs, this would provide a developing thymocyte travelling through a medullary island with the highest likelihood of encountering an mTEC expressing a given TRA against which its antigen receptor could be tested and implies that thymocytes would only need to traverse a limited volume within the thymic medulla in order to be tested against a diverse range of TRAs.

We conclude that reproducible order is evident among the genes that single mTEC express, yet the selection of these genes is indeterminate with respect to biological processes. This degree of randomness may ensure that single TEC express a wide range of TRAs, covering diverse peripheral tissues and auto-antigens, and that expression of a given TRA is spatially dispersed throughout the thymic medulla. In this way, a single thymocyte travelling through the thymic medulla may be given the greatest opportunity of encountering any given self-antigen for the purposes of central tolerance induction.

# Materials and Methods

### Mice

Female C57BL/6, BALB/c and C57BL/6 × BALB/c F1 mice were obtained from Charles River Laboratories (Margate, Kent, UK) and rested for at least 1 week before analysis at 4–5 weeks of age. Mice were housed under specific pathogen-free conditions and according to institutional and UK Home Office regulations.

### Isolation of thymic epithelial cells and preparation for flow cytometry

Thymic epithelial cells were isolated via enzymatic digestion of thymic lobes using Liberase (Roche) and DNaseI (Roche). Cells were counted and stained with anti-CD45 microbeads (Miltenyi Biotec) for 15 min at room temperature, before negative selection using the AutoMACS (Miltenyi Biotec) system in order to enrich for TEC. Samples were then stained for cell surface markers for 20 min at 4°C. For intracellular staining, the Foxp3 Transcription Factor Staining Buffer Kit (eBioscience) was used according to the manufacturer's instructions. Combinations of UEA-1 lectin (Vector Laboratories) labelled in-house with Cy5 and the following antibodies were used to stain the cells: CD45::AF700 (30-F11, BioLegend), EpCAM::PerCPCy5.5 (G8.8, BioLegend), Ly51::PE (6C3, BioLegend), CD80::PECy5 (16-10A1, BioLegend), CD86::PECy7 (GL-1, BioLegend), MHCII::FITC (M5/114.15.2, BioLegend), MHCII::BV421 (M5/114.15.2, BioLegend), GP2::AF488 (2F11-C3, MBL), rat IgG2a κ::AF488 isotype control (eBR2a, eBioscience), TSPAN8::APC

(657909, R&D Systems), rat IgG2a κ::APC isotype control (RTK4530, BioLegend) and desmoglein-3 (DSG3) unlabelled primary antibody (MBL) followed by secondary staining with goat anti-mouse IgG::APC-Cy7 (Abcam), AIRE::AF488 (5H12, eBioscience) and AIRE::AF647 (5H12, eBioscience). For the assessment of cell viability, DAPI or the LIVE/DEAD Fixable Aqua Dead Cell Stain Kit was used (Thermo Fisher Scientific). After staining, cells were acquired and sorted using a FACS Aria III (BD Biosciences) and analysed using FlowJo v10 and GraphPad Prism 7; statistical analyses were performed using a *t*-test with Bonferroni correction for multiple comparisons where appropriate; differences were considered significant if the adjusted *P*-value was ≤ 0.05.

### RNA extraction, reverse transcription and quantitative PCR (RT–qPCR)

RNA was extracted using the RNeasy Micro Kit (Qiagen) and reverse-transcribed using the SensiFAST cDNA Synthesis Kit (Bioline), according to the manufacturer's instructions. qPCR was then performed using the SensiFAST SYBR Hi-Rox Kit (Bioline) and a StepOnePlus real-time PCR instrument (Applied Biosystems), using the following primer pairs (Sigma): *bActin* For 5′GTTCCGATGCCCT GAGGCTC3′, *bActin* Rev 5′CGGATGTCAACGTCACACTTCAT3′; *Gp2* For 5′CAAGAACAGATGCCCAAACCAA3′, *Gp2* Rev 5′AATGGCTGGT CTACTACTGCG3′; and *Tspan8* For 5′TTCAGTCGGAGTTCAAGTGC T3′, *Tspan8* Rev 5′AACGGCCAGTCCAAAAGCAA3′. Data were analysed using GraphPad Prism 7; statistical analyses were performed using the *t*-test with Bonferroni correction for multiple comparisons; differences were considered significant if the adjusted *P*-value was ≤ 0.05.

### Single-cell RNA-sequencing

Cells were FACS sorted into 1.5 ml DNase/RNase-free Eppendorf tubes pre-coated with BSA and containing 150 μl of plain RPMI-1640. Library preparation was carried out on fresh cells directly after FACS sorting using the Chromium Single Cell 3′ V1 Kit or V2 Kit (10× Genomics). The resulting libraries were sequenced on a HiSeq 2500 (Illumina) in High Output mode (paired-end asymmetric 100 bp for read) or a HiSeq4000 (Illumina; paired-end 2 × 75 bp).

### Analysis of single-cell RNA-sequencing results

Libraries were analysed using the Cell Ranger pipeline (10× Genomics) resulting in a Gene-by-Barcode matrix of counts for each biological sample. The secondary analysis was performed using the simpleSingleCell workflow (Lun *et al*, 2016) (Bioconductor). Briefly, each dataset was filtered to remove low-quality libraries. These were defined as cells with a low number of reads (one median absolute deviation (MAD) lower than the median) or features (one MAD lower than the median), or a higher than expected percentage of reads from mitochondrial genes (three MADs higher than the median). The resulting gene-by-cell matrices for each experiment were normalised, and the mnnCorrect algorithm (Haghverdi *et al*, 2018) was used to combine the separate experiments into one corrected meta-experiment.

Graph-based clustering was used to assign the individual mTEC to clusters. First, a shared nearest neighbour (SNN) graph was

generated from the principal components of the normalised gene-by-cell expression matrix. Community structure within the resulting SNN graph was then analysed using the Louvain algorithm (preprint: Blondel *et al*, 2008). This resulted in 15 high-quality clusters of mTEC from the combined meta-experiment.

mTEC were pseudotemporally ordered using two different schemes; firstly, a diffusion map (Haghverdi *et al*, 2015) was used to get a reduced dimensionality representation of the data. That representation was used to order the cells along an inferred trajectory. Next, we used RNAvelocity (La Manno *et al*, 2018) to estimate the time derivative of the gene expression state of our mTEC, thus ordering the mTEC based on spliced and unspliced RNA data captured from each cell.

To enhance the signal from the sparsely expressed TRGs and to prevent widely expressed genes from masking the signal of more sparsely expressed genes, gene module clustering was performed using an adaptation of the TF-IDF (Manning *et al*, 2008) transform. Firstly, a gene-frequency-by-inverse cell-frequency matrix was computed from the normalised gene-by-cell matrix. As this was a co-clustering analysis, only genes that were detected in multiple cells were included in the analysis. The gene frequency portion of the transform was computed as the $\log_2$ of normalised expression or the gene-by-cell expression matrix ($G_f = \log_2$ (C); C is the normalised count matrix of 22,819 features × 6,894 cells). The inverse cell frequency was computed as the weighted average of the inverse frequency of detection of each gene within each subpopulation. That is, for gene $X$ in subset $Y$: if $X$ is detected in 25% of $Y$, then the inverse cell frequency is 4. The five conditions ($TSPAN8^{+/-}$, $GP2^{+/-}$ or unselected) were weighted by their expected contribution to the total mTEC population (Fig 2A; $TSPAN8^+$ 7%, $TSPAN8^-$ 93%, $GP2^+$ 2%, $GP2^-$ 98% and unselected 100% of all mTEC), and the resulting average inverse cell frequency was $\log_{10}$-transformed ($ICF_x = \log10 (\sum_y W_y * N_y/(1 + E_{y,x}))$; $W_y$ is the weight for each subpopulation listed above, $N_y$ is the number of cells in subset $Y$, and $E_{y,x}$ is the number of cells expressing gene $X$ in subset $Y$) before the product of the gene-frequency matrix and inverse cell frequency was computed ($GF\_ICF = G_f * ICF$). The gene-frequency-by-inverse cell-frequency matrix was further reduced to a gene-by-context matrix by using a t-distributed stochastic neighbour embedding (t-SNE) (van der Maaten & Hinton, 2008) to reduce the cosine distance of the first 50 eigenvectors of the gene-frequency-by-inverse cell-frequency matrix (acquired using singular value decomposition) That is: $[d,U,V] = SVD (GF\_ICF)$ (where $d$ is the 50 × 1 vector of singular values, U is the 22,819 × 50 matrix of left singular vectors and V is the 6,894 × 50 matrix of right singular vectors), $D_{cos} = DIST (U, \text{"cosine"})$ is the 22,819 × 22,819 cosine distance matrix for the left singular vectors, and finally $Z = \text{t-SNE} (D_{cos})$ is the 22,819 × 2 matrix of gene context. This reduced dimensionality gene-by-context matrix was then clustered using HDBSCAN (McInnes & Healy, 2017) to spatially select clusters based on density in the reduced dimensionality representation ($GM = HDBSCAN(Z)$ is the gene module assignment from HDBSCAN). This has the benefit of identifying the sets of genes that are repeatedly observed together in the same context (subsets of cells), while simultaneously attenuating the signal from frequently expressed genes unless accompanied by a drastic change in expression level.

Co-expression modules were tested for robustness by repeating the gene module clustering on 100 random subsets of the meta-experiment. The gene modules identified from the random subsets were then compared to determine the overall similarity in the assignment of genes to modules. An adjusted mutual information (AMI) score was used to compare each set of assignments in a pairwise fashion. The clustering algorithm we used either assigns each gene to a module or declares it as noise (unassigned). To compare the clusterings, the AMI was calculated using the unassigned/noise cells as a cluster.

As a secondary analysis to quantify gene co-expression, we calculated the pairwise co-expression frequency of all TRGs within individual mice. This frequency $f(G^X,G^Y)$ was computed as the fraction of cells from a single mouse in which both gene X and gene Y were detected. These frequencies were compared across all mice and to the full meta-experiment (all cells) using a Pearson correlation (Fig 6C). In a restricted version of this analysis, we recomputed these frequencies as above using only mTEC from clusters 3–6 (likely $AIRE^+$ mature mTEC; Appendix Fig S5C).

We examined multiple features of co-expressed genes in order to determine whether a particular feature was able to explain the co-expression patterns that we observe. Firstly, we considered whether each module contains the expected proportion of TRGs or if some module was enriched for TRGs. In this analysis, a Monte Carlo simulation was used to generate the expected number of TRGs in each module and an empirical *P*-value was generated for each observed value. Next, we examined the location of genes within each module to determine whether any modules prefer a particular chromosome. Accordingly, we computed the percentage of genes within a given module that are encoded on each chromosome. Finally, we used a Monte Carlo simulation to determine whether this percentage was more extreme than expected by chance. On a more local scale, we next sought to determine whether co-expressed genes were clustered closer than expected by chance (within chromosomes). To determine this, we computed the full pairwise distance matrix between all pairs of co-expressed genes. We compared the pairwise distance distribution of all genes to only pairs of AIRE-regulated genes or to pairs of genes from the same gene expression module and found very little difference between those distributions and the full distribution for all genes.

## Immunohistochemistry and confocal microscopy

Freshly isolated thymic lobes were frozen in OCT compound (Tissue-Tek) and cryosectioned at a thickness of 8 μm. For immunofluorescence staining, tissue sections were fixed in 10% neutral buffered formalin (Sigma) for 20 min at room temperature and then permeabilised in PBS 0.3% Triton X-100 (Sigma) for 10 min. This was followed by incubation for 1 h at room temperature in blocking buffer consisting of 2% goat serum (Sigma) in PBS 0.1% Triton X-100. For the analysis of $GP2^+$ mTEC, the slides were then stained with a rabbit primary anti-cytokeratin 5 (KRT5) antibody (Covance/BioLegend) diluted 1:500 in blocking buffer for 1 h at 37°C, followed by three 5-min PBS washing steps. Secondary antibody staining with goat anti-rabbit::AF555 diluted 1:500 in PBS was next carried out for 30 min at 37°C, followed by three 5-min PBS washing steps. A third staining step with a 1:200 dilution of an anti-GP2 antibody directly conjugated to AF488 (MBL, 2F11-C3) or an isotype control (rat IgG2a κ::AF488 isotype control, eBR2a,

eBioscience) was subsequently performed for 1 h at 37°C. After washing three times with PBS, the slides were stained using 500 ng/ml DAPI (Sigma) diluted in methanol (VWR), washed once in PBS and mounted using ProLong Gold Antifade mounting medium (Life Technologies). For the analysis of TSPAN8$^+$ mTEC, primary antibody staining was first performed using a guinea pig anti-cytokeratin 14 (KRT14) antibody (Abcam) diluted 1:200 and a rat anti-TSPAN8 antibody (R&D, MAB6524) diluted 1:100 in blocking buffer for 1 h at 37°C. After three 5-min washes with PBS, secondary antibody staining with goat anti-guinea pig::AF488 (BioLegend) diluted 1:500 and goat anti-rat::AF647 (BioLegend) diluted 1:500 in PBS was carried out for 30 min at 37°C, followed by three 5-min PBS washing steps. Secondary antibody only staining was used as a control for TSPAN8. After secondary antibody staining, slides were washed three times with PBS, stained with DAPI and mounted as above. After washing as above, the slides were stained using 500 ng/ml DAPI (Sigma) diluted in methanol (VWR), washed once in PBS and mounted using ProLong Gold Antifade mounting medium (Life Technologies). Imaging was performed on an LSM 780 (for GP2 analyses) or 880 (for TSPAN8 analyses) inverted confocal microscope (Ziess) and analysed using Fiji (Schindelin *et al*, 2012).

### Spatial analysis of TSPAN8$^+$ and GP2$^+$ cell distributions

Four representative $1,980 \times 1,980$ pixel microscope images co-stained with DAPI, GP2 and KRT5 were used to identify GP2$^+$ mTEC and the medullary region within thymic images. Ten representative $512 \times 512$ pixel microscope images co-stained with DAPI, TSPAN8 and KRT14 were used to identify TSPAN8$^+$ mTEC and the medullary region within thymic images. Each colour image was opened and processes using the EBImage package (R; Pau *et al*, 2010). KRT5/KRT14 layers were processed with a low pass Gaussian filter; then, thresholded images were eroded and dilated to obtain a mask of the region covered by mTEC. A similar process was used to identify GP2$^+$/TSPAN8$^+$ mTEC on the GP2$^+$/TSPAN8$^+$ layer, an additional stage of watershed processing completed the labels of individual GP2$^+$/TSPAN8$^+$ mTEC. The moment of each GP2$^+$/TSPAN8$^+$ feature was computed and validated by visual inspection, and these positions were used in the spatial analysis. To determine whether the GP2$^+$ mTEC were clustered or randomly dispersed within the medullary region, the nearest neighbour distance distribution function (G(r): spatstat package R) was calculated for the point pattern derived from the moments of the GP2$^+$/TSPAN8$^+$ mTEC within the space classified as medulla by the KRT5$^+$/KRT14$^+$ mask generated above.

## Data availability

These data are available through ArrayExpress under the ID E-MTAB-8105 (http://www.ebi.ac.uk/arrayexpress/experiments/E-MTAB-8105/).

**Expanded View** for this article is available online.

## Acknowledgements
This work was supported by the Wellcome Trust (109032/Z/15/Z, 105045/Z/14/Z) and the MRC (MC_UU_00007/15 to CPP).

## Author contributions
FD, SM, JB-G, GH and CPP designed the experiments. FD and SM conducted the experiments; LC helped with 10× library preparation. FD and JB-G analysed the data and produced the figures. FD, JB-G, GH and CPP contributed to writing the manuscript.

## Conflict of interest
The authors declare that they have no conflict of interest.

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
