## [Review Process File · The EMBO Journal]

Biologically indeterminate yet ordered promiscuous gene expression in single medullary thymic epithelial cells

F. Dhalla, J. Baran-Gale, S. Maio, L. Chappell, G. Holländer, C.P. Ponting

Review timeline:

Submission date:	18 February 2019
Editorial Decision:	9 April 2019
Revision received:	27 June 2019
Editorial Decision:	22 August 2019
Revision received:	20 September 2019
Accepted:	25 September 2019

Editor: Karin Dumstrei

Transaction Report:

1st Editorial Decision

9 April 2019

Thank you for submitting your manuscript to The EMBO Journal. Your study has now been seen by two referees and their comments are provided below.

As you can see, the referees find the analysis well done, interesting and insightful. Most of the points raised should be fairly straightforward to address. I would therefore like to invite you to submit a revised version. Both referees also indicate that the analysis remains a bit descriptive and that some further follow up analysis would be good. I like the suggestion of referee #2 to look if co-regulated genes also physically co-localise like via chromatin looping. I don't know how feasible this is to do and I am happy to discuss further.

When preparing your letter of response to the referees' comments, please bear in mind that this will form part of the Review Process File, and will therefore be available online to the community. For more details on our Transparent Editorial Process, please visit our website: http://emboj.embopress.org/about#Transparent_Process

Thank you for the opportunity to consider your work for publication. I look forward to your revision.

REFeree REPORTS

Referee #1:

In this manuscript Dhalla et al. performed single cell RNA sequencing (scRNA-Seq) of medullary thymic epithelial cell (mTEC) compartment in order to examine its heterogeneity and to better understand the rules governing the phenomenon of promiscuous gene expression (PGE), which is the key hallmark of mTEC.

Although scRNA-seq analysis of mTEC has already been done in the past by several independent groups (Sansom et al, Berennecke et al, Meredith et al, Miagara et al, Bornstein et al), the study by Dhala et al represents probably the most comprehensive recourse with the largest coverage of cells and depth of sequencing so far. Importantly, this study largely validates findings by Bornstein et al and Miagara et al, including the identification of thymic tuft cells, Ccl21/Pdnp+ mTEC and Krt10/Ivl+ mTEC. In addition, however, by combining unbiased scRNA-seq analysis together with scRNA-seq analysis of cells expressing either TSPAN8 or GP2 (i.e. two different Aire-dependent tissue restricted antigen genes), the authors were also able to highlight additional mTEC subsets that may have been overlooked in previous studies (e.g. Ccl6+ mTEC).

In addition, unlike previous studies by Sansom et al, Bernnecke et al and Meredith et al, who studied the phenomenon of PGE exclusively in the in the MHCII-high mTEC subset, Dhala et al performed a thorough analysis of PGE at a single cell level throughout all mTEC. This is in particular essential for understanding the rules of PGE not only in Aire-expressing mTEC, but also in their progeny.

This is probably the most novel aspect of this study. Specifically, based on their data, the authors try to challenge previous findings that suggested that PGE is highly stochastic in nature and rather argue that PGE is ordered and reflects specific developmental stages of mTEC.

Finally, the authors also describe the potential developmental trajectories of mTEC. It should be stressed, however, that the authors do not use any independent method of validation (e.g. lineage tracing, ablation of particular population etc.) to support their data. Thus, although the proposed trajectories seem logical, they are largely speculative.

Although the work is in general well done and analyzed, the main weakness of the study is that it is mainly descriptive. Nevertheless, the study represents a very impressive and important resource of data that may be very useful for subsequent studies in this field and thus may be suitable for publication in EMBO J.

Major points:

The authors show that mature mTEC clusters express distinct TRGs, and that this is reproducible in different mice. These data prompt the authors to question the stochastic nature of PGE and rather propose that PGE is regulated by an ordered mechanism. Although this statement may be true, it may be inaccurate as it may concern only the post-Aire mTEC. Specifically, it is now well established that Aire+ mTEC do not represent a final stage of mTEC development but rather continue to differentiate into distinct terminally differentiated subsets that highly resemble epidermal keratinocytes or tuft cells (and likely other epithelial cell types) in their molecular and morphological characteristics. Therefore, it is not surprising that the authors find ordered rules for co-expression of specific gene signatures in these clusters, as they logically follow the mTEC differentiation process. Correspondingly, it is not surprising that such patterns are also observed for Tspan8 and Gp2 (the two model TRG used in this study), as they are predominantly expressed by post-Aire cells (e.g. the initial studies by Kyewski group clearly marked Tspan8 as a marker that is co-expressed with involucrin, indicating that Tspan8 is most likely found in terminally differentiated mTEC).

Therefore, while ordered developmental rules may determine the co-expression pattern in the terminally differentiated subsets, the rules controlling PGE in the major cloud of Aire-expressing cells may be very different and still be highly stochastic, as had been proposed by some of the initial studies. This issue needs to be resolved further by thorough analysis of all TRG in the major Aire-expressing cloud. If PGE is also ordered and not stochastic in the Aire+mTEC, why previous studies based on single cell sequencing got to a different conclusion? This should be discussed and explained in detail, otherwise this study will bring more confusion than insights to the field.

Minor points

- 1) The abstract is not very informative and does a great injustice to the study.
- 2) Authors should refrain from using the claim, that this is the largest single-cell RNA-seq dataset. This might be true nowadays, however doubtfully for a longer time period.

- 3) In the introduction, the authors describe four alternative hypotheses for how PGE may be regulated. Given the complexity of these processes, it might be better to present these 4 models in a graphical illustration in a supplemental figure.
- 4) Tspan and Gp2 should be already explained in introduction section.
- 5) Quote: "The robust identification of 15 distinct mTEC clusters suggested an ordered process that selects TRGs to be co-expressed within single mTEC" - It is not clear how the authors came to the conclusion in this sentence?
- 6) The FACS plots showing potential co-expression or no co-expression of TSPAN8 and GP2 should be present.
- 7) In Figure 1d and 2- it's hard to see whether the Gp2 preferred or Tspan8 preferred clusters are indeed preferred, some statistics would make it more obvious/convincing. For example, in 2a it seems that there are many green cells also in the left upper cluster, and in 2b it seems that there are many brown cells also in the right upper cluster. It would be nice to have some kind of statistic saying that this is less than expected (i.e % of brown cells in the right upper cluster is lower than their % in the whole population) and vice versa for the preferred clusters...
- 8) GP2+ mTEC are part of very different clusters in C57Bl/6 mice and Balb/c x C57Bl/6 hybrids. Obviously, the protein expression of this TRAs persist and this can serve as kind of lineage tracing. However, the clusters formed by GP2+ TECs are rather different in both different mouse strains. Can authors address this issue. Moreover, clustering of Tspan8+ TECs is shown only for C57Bl/6. How the situation looks like in hybrids and Balb/c?
- 9) Although the authors claim that the expression of GP2 and Tspan8 is ordered, the distribution of GP2/Tspan8 positive cells looks semi-stochastic as both antigens span across virtually all major clusters, though they seem to be more concentrated in specific regions.
- 10) Could the authors also provide some kind of correlation of the protein expression of Tspan8 and Gp2 vs their corresponding mRNA
- 11) Fig 3D is very helpful as it elegantly shows the expression of Aire dependent TRA genes across the mTEC subsets. This helps to strengthen the data presented in the developmental trajectory analysis, as it highlights cells with and without Aire footprint. To gain a more quantitative picture, it would be very useful to include #TRG/cell in each specific cluster. In addition, could the authors also include distribution of Aire independent TRG.
- 12) Trajectories - the first 2 seem to make sense but the last seems a bit unexpected as it suggests that Aire expressing cells downregulate Aire expression and become progenitors? Could the authors discuss this further and/or provide some experimental backing for this hypothesis?
- 13) The conclusions regarding localization of TRG is a bit weak, as it is based on microscopic analysis of only one gene (Gp2). More TRGs should be shown to generalize this statement
- 14) The satellite clusters are poorly characterized - very little info is provided about the specific genes that characterize each of these clusters. Therefore, it would be highly appreciated to provide more molecular details for each of the clusters in suppl info
- 15) The gene module analysis is very interesting, but should be better explained and more transparent. What genes comprise the specific module should be provided in suppl info.

Referee #2:

mTEC as a whole showed promiscuous gene expression (PGE) which is crucial for central T-cell tolerance. To unveil the molecular basis of PGE, the present report describes the transcriptome of 6,894 single mTEC that were either unselected or that expressed either Tetraspanin 8 or Glycoprotein2 and that originated from C57BL/6, BALB/c and C57BL/6 x BALB/c female mice. A median of 1,830 genes was detected per cell of which some were AIRE-regulated TRGs. t-SNE visualization indicated that mTEC subsets expressing a particular TRA expressed a transcriptome that distinguish them from most other mTEC subsets. 15 mTEC populations were disentangled, including some corresponding to maturational stages. Trajectory analyses corroborated some of the postulated cluster precursor-product relationships. 14,861 genes were assigned to 50 modules. Importantly, frequently expressed genes do not contribute substantially to the module identities. Moreover, the gene co-expression patterns highlighted in most of the 50 gene modules were independent of their location on a given chromosome. Furthermore, such patterns of gene expression were not consistent with a transcriptional program already found in a peripheral tissue. Glycoprotein 2-expressing mTEC were randomly distribution within the medulla. The thorough analysis of this large dataset permits to exclude several molecular models of PGE expression in single medullary thymic epithelial cells and goes beyond a recent study by Bornstein et al. (2018) and in which only

four classes of mTEC were identified using single cell mRNA seq. It remains however a rather phenomenological study. It will have been nice to use available technologies to tackle whether co-regulated genes are physically co-located on chromatin by virtue of chromatin looping.

Specific questions

1/ Only half of the mTEC in clusters 10 and 14 were positive for the corresponding TRA. Is it due to RNA sampling issues or improper sorting ?

1st Revision - authors' response

27 June 2019

Referee #1:

In this manuscript Dhalla et al. performed single cell RNA sequencing (scRNA-Seq) of medullary thymic epithelial cell (mTEC) compartment in order to examine its heterogeneity and to better understand the rules governing the phenomenon of promiscuous gene expression (PGE), which is the key hallmark of mTEC.

Although scRNA-seq analysis of mTEC has already been done in the past by several independent groups (Sansom et al, Berenke et al, Meredith et al, Miagara et al, Bornstein et al), the study by Dhalla et al represents probably the most comprehensive recourse with the largest coverage of cells and depth of sequencing so far. Importantly, this study largely validates findings by Bornstein et al and Miagara et al, including the identification of thymic tuft cells, Ccl21/Pdpn+ mTEC and Krt10/Ivl+ mTEC. In addition, however, by combining unbiased scRNA-seq analysis together with scRNA-seq analysis of cells expressing either TSPAN8 or GP2 (i.e. two different Aire-dependent tissue restricted antigen genes), the authors were also able to highlight additional mTEC subsets that may have been overlooked in previous studies (e.g. Ccl6+ mTEC).

In addition, unlike previous studies by Sansom et al, Berenke et al and Meredith et al, who studied the phenomenon of PGE exclusively in the MHCII-high mTEC subset, Dhalla et al performed a thorough analysis of PGE at a single cell level throughout all mTEC. This is in particular essential for understanding the rules of PGE not only in Aire-expressing mTEC, but also in their progeny. This is probably the most novel aspect of this study. Specifically, based on their data, the authors try to challenge previous findings that suggested that PGE is highly stochastic in nature and rather argue that PGE is ordered and reflects specific developmental stages of mTEC.

Finally, the authors also describe the potential developmental trajectories of mTEC. It should be stressed, however, that the authors do not use any independent method of validation (e.g. lineage tracing, ablation of particular population etc.) to support their data. Thus, although the proposed trajectories seem logical, they are largely speculative.

Although the work is in general well done and analyzed, the main weakness of the study is that it is mainly descriptive. Nevertheless, the study represents a very impressive and important resource of data that may be very useful for subsequent studies in this field and thus may be suitable for publication in EMBO J.

Major points:

The authors show that mature mTEC clusters express distinct TRGs, and that this is reproducible in different mice. These data prompt the authors to question the stochastic nature of PGE and rather propose that PGE is regulated by an ordered mechanism. Although this statement may be true, it may be inaccurate as it may concern only the post-Aire mTEC. Specifically, it is now well established that Aire+ mTEC do not represent a final stage of mTEC development but rather continue to differentiate into distinct terminally differentiated subsets that highly resemble epidermal keratinocytes or tuft cells (and likely other epithelial cell types) in their molecular and morphological characteristics. Therefore, it is not surprising that the authors find ordered rules for co-expression of specific gene signatures in these clusters, as they logically follow the mTEC differentiation process. Correspondingly, it is not surprising that such patterns are also observed for Tspan8 and Gp2 (the two model TRG used in this study), as they are predominantly expressed by post-Aire cells (e.g. the initial studies by Kyewski group clearly marked Tspan8 as a marker that is co-expressed with involucrin, indicating that Tspan8 is most likely found in terminally differentiated mTEC).

Therefore, while ordered developmental rules may determine the co-expression pattern in the terminally differentiated subsets, the rules controlling PGE in the major cloud of Aire-expressing

cells may be very different and still be highly stochastic, as had been proposed by some of the initial studies. **This issue needs to be resolved further by thorough analysis of all TRG in the major Aire-expressing cloud. If PGE is also ordered and not stochastic in the Aire+mTEC, why previous studies based on single cell sequencing got to a different conclusion? This should be discussed and explained in detail, otherwise this study will bring more confusion than insights to the field.**

We thank the reviewer for this helpful comment. To clarify the issue relating to co-expression of TRGs we repeated the analysis previously represented in Figure 6c for all mTEC, except now only using those mTEC from the likely Aire⁺ clusters (3-6) and excluding the likely terminally differentiated clusters (7-9). In this restricted analysis, we compared the co-expression frequency of all pairs of tissue restricted genes (both AIRE-regulated and AIRE-independent) and found that co-expression of TRGs remains highly similar across mice (mean Pearson correlation of 0.69 compared to 0.75 across all cells; **Figure S5c**). Previous studies concluded that TRG co-expression is ordered or stochastic. The main differences between our study and these earlier studies are: (1) that we investigated many more mTEC and (2) we have analysed co-expression by computing the full pairwise co-expression frequency across all TRGs and considering these results in aggregate. Whereas previous studies have investigated few mTEC (201 across two pairs of WT and Aire KO mice: Meredith et al 2015; 141 Aire⁺ mTEC: Sansom et al. 2014; 203 MHCII^{hi} TEC pooled from 5-20 mice: Brennecke et al. 2015), we acquired a dataset of 6,894 mTEC from multiple mice that was then used to investigate TRG co-expression.

Figure S5c: Pearson correlation of co-expression frequency for pairs of TRGs (AIRE-regulated and AIRE-independent) between individual mice or all mice pooled together (see Table S1 for mouse identifiers) from mature mTEC clusters only (clusters 3-6).

We also made the following modification to the text:

“Next, we sought to determine the variability of TRG co-expression between individual mice (of either the same or different genetic backgrounds). For each pair of TRGs, the fraction of mTEC expressing both TRGs was calculated per mouse, these fractions were then compared across all mice in our dataset. The mean Pearson correlation of these fractions across all mouse pairs was high (average value of 0.75; Figure 6c). **To further investigate whether the TRG co-expression frequencies were driven by the post-AIRE mTEC we recomputed these frequencies as above using only mTEC from clusters 3-6 (likely AIRE⁺ mature mTEC; Figure S5c). This restricted analysis showed that TRG co-expression frequencies remained highly correlated (mean Pearson correlation of 0.69).** Our findings thus demonstrate that sets of TRGs were repeatedly co-expressed in individual mTEC and that this TRG co-expression was replicated both in random subsets of mTEC and across different mice.”

Minor points

1) The abstract is not very informative and does a great injustice to the study.

We have modified the abstract as follows:

“To induce central T-cell tolerance, medullary thymic epithelial cells (mTEC) collectively express most protein coding genes, thereby presenting an extensive library of tissue-restricted antigens (TRAs). Whether this process of promiscuous gene expression (PGE) is stochastic or coordinated is, however, unknown. To resolve this, we sequenced the transcriptomes of 6,894 single mTEC, enriching for 1,795 rare cells expressing either of two TRAs, TSPAN8 or GP2. Transcriptional heterogeneity allowed partitioning of mTEC into 15 reproducible subpopulations representing distinct maturational trajectories, stages and subtypes. Unexpectedly, 50 groups of genes were robustly defined each showing patterns of co-expression within individual cells, yet most of these could not be explained by chromosomal location, biological pathway, or tissue specificity. Further, TSPAN8+ and GP2+ mTEC locations were randomly dispersed within medullary islands. In summary, although PGE exhibits ordered co-expression, the mechanism underlying this order remains biologically indeterminate. Tissue-independent TRA expression in co-expression clusters and the random spatial distribution of TRAs within the thymic medulla likely enhance the presentation of a diverse catalog of antigens and its encounter by passing thymocytes, whilst simultaneously maintaining mTEC identity throughout PGE.”

2) Authors should refrain from using the claim, that this is the largest single-cell RNA-seq dataset. This might be true nowadays, however doubtfully for a longer time period.

Agreed. We have now removed any claim within the first results section that this is the “largest single-cell RNA-seq dataset investigating PGE in mTEC”.

3) In the introduction, the authors describe four alternative hypotheses for how PGE may be regulated. Given the complexity of these processes, it might be better to present these 4 models in a graphical illustration in a supplemental figure.

We thank the reviewer for this suggestion. We now provide a figure depicting these alternatives and also refer to it in the text.

Figure 1: Processes that could regulate TRG co-expression. (1) Stochastic: Gene co-expression is a fully stochastic process; (2) Maturation stage: Co-expression is driven by mTEC maturation stages; (3) Re-use: Co-expression is driven by re-use of existing tissue restricted programs of gene expression; (4) Physical co-location: Co-expressed genes are in close physical proximity.

4) Tspan and Gp2 should be already explained in introduction section.

We thank the reviewer for catching this. We have resolved this oversight by defining the acronyms TSPAN8 and GP2 in the final paragraph of the Introduction. In addition, a sentence describing the tissue restricted expression of these antigens in the periphery has been moved from the first results section to the Introduction. The final part of the introduction now reads:

“...the narrow range contained two sets of mTEC that are rare in expressing Tetraspanin 8 (TSPAN8) or Glycoprotein2 (GP2), two AIRE-regulated TRAs. TSPAN8 is expressed in the gastrointestinal tract and several carcinomas and GP2 is expressed in the pancreas and gastrointestinal tract; loss of tolerance to GP2 is associated with Crohn’s disease and primary sclerosing cholangitis”

5) Quote: "The robust identification of 15 distinct mTEC clusters suggested an ordered process that selects TRGs to be co-expressed within single mTEC" - It is not clear how the authors came to the conclusion in this sentence?

We apologise for the previous lack of clarity. This sentence has been changed to: “The robust identification of 15 distinct mTEC clusters, which were largely preserved when clustering was performed using only TRGs (Figure S5d), suggested an ordered process that selects TRGs to be co-expressed within single mTEC.”

6) The FACS plots showing potential co-expression or no co-expression of TSPAN8 and GP2 should be present.

To address this, Figure 2a has been altered and now contains a FACS plot showing TSPAN8 vs GP2 expression within mTEC. Furthermore, the bar graph now also shows the percentage of mTEC that co-express TSPAN8 and GP2 proteins.

Figure 2a: mTEC promiscuously expressing TSPAN8 and/or GP2 (upper right panel/red) on their cell surface can be identified by flow cytometry. mTEC were identified as CD45-EpCAM+Ly51- (Fig. S1) and the gates for TSPAN8/GP2 were set against isotype control antibodies (left panels/grey). Lower right panel: Bar graph showing mean frequency (+/- sd) of TSPAN8+, GP2+, and TSPAN8+GP2+ cells within total mTEC; results represent pooled data from 3 (TSPAN8+), 4 (GP2+) and 2 (TSPAN8+GP2+) independent experiments each containing 3 individual mice.

7) In Figure 1d and 2- it's hard to see whether the Gp2 preferred or Tspan8 preferred clusters are indeed preferred, some statistics would make it more obvious/convincing. For example, in 2a it seems that there are many green cells also in the left upper cluster, and in 2b it seems that there are many brown cells also in the right upper cluster. It would be nice to have some kind of statistic saying that this is less than expected (i.e. % of brown cells in the right upper cluster is lower than their % in the whole population) and vice versa for the preferred clusters...

To address this issue, we now include the percentage of all mTEC from a given sample that are assigned to each cluster and compared this to the percentage of Unselected mTEC. In Figure S4b, we show that several clusters are enriched or depleted for either TSPAN8 or GP2+/- mTEC. Additionally, we have modified the text to include p-values and references to this supplemental figure when introducing each cluster.

Figure S4b: Bar plot showing the % of cells from each sample that fall within each cluster. Bars are coloured by FACS sort condition as listed in (a), clusters that have a significant ($p < 0.05$; Wilcoxon test) difference in mean frequency as compared to unselected mTEC are annotated with a *. (top) TSPAN8+/- vs unselected mTEC (middle) GP2+/- vs unselected mTEC (bottom) Comparison of C57BL/6 vs BALB/c unselected mTEC.

8) GP2+ mTEC are part of very different clusters in C57BL/6 mice and Balb/c x C57BL/6 hybrids. Obviously, the protein expression of this TRAs persist and this can serve as kind of lineage tracing. However, the clusters formed by GP2+ TECs are rather different in both different mouse strains. Can authors address this issue. Moreover, clustering of Tspan8+ TECs is shown only for C57BL/6. How the situation looks like in hybrids and Balb/c?

We thank the reviewer for this comment. To clarify: we did not perform sequencing of TSPAN8+ mTEC derived from BALB/c or F1 hybrids and hence are unable to show how TSPAN8+ mTEC cluster for these mouse strains. However, we did assess the assignment of Unselected mTEC from either C57BL/6 or BALB/c and found no difference in the levels (included in Figure S4b directly above). Unfortunately, mTEC from only two mice were included in these data. Consequently, we cannot rule out the possibility that we lack the power to detect significant differences in cluster composition between these strains.

9) Although the authors claim that the expression of GP2 and Tspan8 is ordered, the distribution of GP2/Tspan8 positive cells looks semi-stochastic as both antigens span across virtually all major clusters, though they seem to be more concentrated in specific regions.

We feel that overall our data is supportive of TRG expression in general (i.e. not limited to TSPAN8 and GP2) having order. This is particularly evident in the gene module clustering analysis, which revealed 50 robustly defined co-expression groups, the majority of which contained TRG co-expression sets. That TRG expression has order is, of course, distinct from it being a fully ordered process. Therefore, the text has been carefully reviewed and some of the text has been altered to ensure that this notion is accurately pitched. For example:

“Within the central body, the majority of mTEC fell along a manifold characterised by a transition from predominantly TSPAN8- or GP2- mTEC at the lower right pole, to TSPAN8+ or GP2+ mTEC at the upper left pole (Figure 2d, Figure 3). TSPAN8+ and GP2+ mTEC each contributed to distinct satellite clusters (Figure 3 green and brown arrows in panels a, b, d).” **has been changed to** “Within the central body, the majority of mTEC fell along a manifold characterised by a transition from predominantly TSPAN8- or GP2- mTEC at the lower right pole, to **predominantly** TSPAN8+ or GP2+ mTEC at the upper left pole (Figure 2d, Figure 3). TSPAN8+ and GP2+ mTEC each **preferred** distinct satellite clusters (Figure 3 green and brown arrows in panels a, b, d).”

“These observations argue for an uneven expression of TRGs across mTEC subpopulations, implying that satellite clusters show preference for expression of particular gene subsets and providing additional evidence against a Type 1 process (TRG expression is stochastic), and in favour of a Type 2 process (different maturational stages or classes of mTEC activate TRG expression differentially).” **has been changed to** “These observations argue for an uneven expression of TRGs across mTEC subpopulations, implying that satellite clusters show preference for expression of particular gene subsets and providing additional evidence against a Type 1 process (TRG expression is **entirely** stochastic), and in favour of a Type 2 process (different maturational stages or classes of mTEC activate TRG expression differentially).”

“In summary, 50 gene co-expression modules were identified, half of which were largely driven by TRG co-expression patterns reproducible in different mice. This again argues against a stochastic mechanism for TRG expression within single mTEC (Introduction).” **has been changed to** “In summary, 50 gene co-expression modules were identified, half of which were largely driven by TRG co-expression patterns reproducible in different mice. This again argues against **an entirely** stochastic mechanism for TRG expression within single mTEC (Introduction).”

“Whilst gene expression in mTEC is ordered, gene membership in co-expression clusters is biologically indeterminate” **has been changed to** “Whilst gene expression in mTEC **has order**, gene membership in co-expression clusters is biologically indeterminate”

10) Could the authors also provide some kind of correlation of the protein expression of Tspan8 and Gp2 vs their corresponding mRNA

Figure 2b shows that protein expression correlates with mRNA expression of TSPAN8 and GP2 at the bulk level. Comparing Figures 2d and 3a, b and c, which show the cells' FACS phenotype and hence the protein expression status for TSPAN8 and GP2 with Figures 4e and f, which show the cells mRNA expression level for the two TRGs, should additionally provide an impression of the correlation between protein and mRNA expression of TSPAN8 and GP2 at the single cell level, although detection of the latter will of course be limited by capture efficiency.

11) Fig 3D is very helpful as it elegantly shows the expression of Aire dependent TRA genes across the mTEC subsets. This helps to strengthen the data presented in the developmental trajectory analysis, as it highlights cells with and without Aire footprint. To gain a more quantitative picture, it would be very useful to include #TRG/cell in each specific cluster. In addition, could the authors also include distribution of Aire independent TRG.

To address this, we have now included panels a and b in Figure S5. These panels display the number of TRGs (AIRE-dependent, AIRE-enhanced and AIRE-independent TRGs) overlaid over the tSNE visualisation (Figure S5a) or as a histogram separated by cluster number (Figure S5b).

Figure S5a-b. Contribution to mTEC clusters. (a) Log10 of the number (#) of AIRE-dependent, AIRE-enhanced and AIRE-independent TRGs expressed per cell visualised on a tSNE plot. (b) Histogram showing the number (#) of AIRE-dependent, AIRE-enhanced and AIRE-independent TRGs expressed per cell within each cluster.

12) Trajectories - the first 2 seem to make sense but the last seems a bit unexpected as it suggests that Aire expressing cells downregulate Aire expression and become progenitors? Could the authors discuss this further and/or provide some experimental backing for this hypothesis?

We agree with the referee that it would not make sense and, indeed, would be out of keeping with published data that Aire expression in TEC is bi-phasic and not detected in progenitors. We used two orthogonal approaches to order the mTEC in pseudotime and have come to our conclusions having considered the output of both analyses:

“Taken together, these results suggest that proliferating mTEC in cluster 3 and Aire+ mTEC in clusters 4-6 originated from the Aire-Cd80-Cd86- mTEC in cluster 2. The Aire-Cd80-Cd86- mTEC from clusters 7-8 appeared to derive from mature mTEC of clusters 5-6 (Yano et al, 2008; Michel et al, 2017; Wang et al, 2012) and were transcriptionally distinct from the Aire-Cd80-Cd86- cells in clusters 1 and 2. **Consequently, we propose that clusters 1 and 2 represent pre-AIRE mTEC (distinguished by Ccl21a and Pdpn expression) while clusters 7 and 8 represent post-AIRE mTEC (distinguished by Iv1, K10 and Spink5 expression).**”

13) The conclusions regarding localization of TRG is a bit weak, as it is based on microscopic analysis of only one gene (Gp2). More TRGs should be shown to generalize this statement

We have now been able to stain for TSPAN8 and include this in our analyses. Interestingly, the spatial distribution of TSPAN8 within medullary islands was also found to be random. Despite best efforts, we were unable, within the time frame available, to undertake spatial analysis for further TRAs.

The discussion has been altered so as not to overstate the conclusions drawn and now reads:

“Furthermore, the spatial locations of TSPAN8+ and GP2+ mTEC are randomly dispersed across the thymic medulla. **Should the observed random spatial distribution of TSPAN8 and GP2 presentation be generalisable to other TRAs**, this would provide a developing thymocyte travelling through a medullary island with the highest likelihood of encountering an mTEC expressing a given TRA against which its antigen receptor could be tested and implies that thymocytes would only need to traverse a limited volume within the thymic medulla in order to be tested against a diverse range of TRAs”

14) The satellite clusters are poorly characterized - very little info is provided about the specific genes that characterize each of these clusters. Therefore, it would be highly appreciated to provide more molecular details for each of the clusters in suppl info

We thank the reviewer for this comment and have now provided Table S2 to contribute this information.

15) The gene module analysis is very interesting, but should be better explained and more transparent. What genes comprise the specific module should be provided in suppl info.

Agreed. We now add Table S3 to provide this information. Additionally, we have expanded the methods section as follows:

“To enhance the signal from the sparsely expressed TRGs and to prevent widely expressed genes from masking the signal of more sparsely expressed genes, gene module clustering was performed using an adaptation of the TF-IDF(Manning *et al*, 2008) transform. Firstly, a gene-frequency-by-inverse cell-frequency matrix was computed from the normalised gene-by-cell matrix. As this was a co-clustering analysis, only genes that were detected in multiple cells were included in the analysis. The gene frequency portion of the transform was computed as the log₂ of normalised expression or the gene-by-cell expression matrix ($G_f = \log_2(C)$; **C is the normalized count matrix of 22,819 features x 6,894 cells**). The inverse cell-frequency was computed as the weighted average of the inverse frequency of detection of each gene within each subpopulation. That is, for gene *X* in subset *Y*: if *X* is detected in 25% of *Y* then the inverse cell-frequency is 4. The five conditions (TSPAN8+/-, GP2+/- or Unselected) were weighted by their expected contribution to the total mTEC population (Figure 2a; **TSPAN8+ 7%, TSPAN8- 93%, GP2+ 2%, GP2- 98% and unselected 100% of all mTEC**) and the resulting average inverse cell-frequency was log₁₀ transformed ($ICF_x = \log_{10}(\sum_y W_y * N_y / (1+E_{y,x}))$; **W_y is the weight for each subpopulation listed above, N_y is the number of cells in subset *Y*, $E_{y,x}$ is the number of cells expressing gene *X* in subset *Y***) before the product of the gene-frequency matrix and inverse cell-frequency were computed ($GF_ICF = G_f * ICF$). The gene-frequency-by-inverse cell-frequency matrix was further reduced to a gene-by-context matrix by using a t-distributed stochastic neighbour embedding (t-SNE)(Maaten & Hinton, 2008) to reduce the cosine distance of the first 50 eigenvectors of the gene-frequency-by-inverse cell-frequency matrix (acquired using singular value decomposition) **That is: $[d,U,V] = SVD(GF_ICF)$ (where *d* is the 50x1 vector of singular values, *U* is the 22,819x50 matrix of left singular vectors and *V* is the 6,894x50 matrix of right singular vectors), $D_{cos} = DIST(U, 'cosine')$ is the 22,819x22,819 cosine distance matrix for the left singular vectors, and finally $Z = tSNE(D_{cos})$ is the 22,819x2 matrix of gene context**. This reduced dimensionality gene-by-context matrix was then clustered using HDBSCAN(McInnes & Healy, 2017) to spatially select clusters based on density in the reduced dimensionality representation (**$GM = HDBSCAN(Z)$ is the gene module assignment from HDBSCAN**). This has the benefit of identifying the sets of genes that are repeatedly observed together in the same context (subsets of cells), while simultaneously attenuating the signal from frequently expressed genes unless accompanied by a drastic change in expression level.”

Referee #2:

mTEC as a whole showed promiscuous gene expression (PGE) which is crucial for central T-cell tolerance. To unveil the molecular basis of PGE, the present report describes the transcriptome of 6,894 single mTEC that were either unselected or that expressed either Tetraspanin 8 or Glycoprotein2 and that originated from C57BL/6, BALB/c and C57BL/6 x BALB/c female mice. A median of 1,830 genes was detected per cell of which some were AIRE-regulated TRGs. t-SNE visualization indicated that mTEC subsets expressing a particular TRA expressed a transcriptome

that distinguish them from most other mTEC subsets. 15 mTEC populations were disentangled, including some corresponding to maturational stages. Trajectory analyses corroborated some of the postulated cluster precursor-product relationships. 14,861 genes were assigned to 50 modules. Importantly, frequently expressed genes do not contribute substantially to the module identities. Moreover, the gene co-expression patterns highlighted in most of the 50 gene modules were independent of their location on a given chromosome. Furthermore, such patterns of gene expression were not consistent with a transcriptional program already found in a peripheral tissue. Glycoprotein 2-expressing mTEC were randomly distributed within the medulla. The thorough analysis of this large dataset permits to exclude several molecular models of PGE expression in single medullary thymic epithelial cells and goes beyond a recent study by Bornstein et al. (2018) and in which only four classes of mTEC were identified using single cell mRNA seq. It remains however a rather phenomenological study. **It will have been nice to use available technologies to tackle whether co-regulated genes are physically co-located on chromatin by virtue of chromatin looping.**

Evidence for this possibility exists from previous studies. Pinto *et al.* used FISH to show that co-expressed TRGs are co-localised within the same nuclear subdomains in mTEC. In addition, Bansal *et al.* found AIRE to localise to and activate superenhancers in mTEC, which are thought to activate gene expression by looping to transcriptional start sites. The text has been altered to describe these published observations more explicitly **from** “By contrast, our data provided evidence that different maturational stages or classes of mTEC activate TRG expression differentially (Type 2) and our data do not exclude the possibility that TRGs are physically co-located on chromatin by virtue of chromatin looping (Type 4b), as has been suggested (Bansal et al, 2017; Pinto et al, 2013).” **to** “By contrast, our data provided evidence that different maturational stages or classes of mTEC activate TRG expression differentially (Type 2, Figure 1) and our data do not exclude the possibility that TRGs are physically co-located on chromatin by virtue of chromatin looping (Type 4b, Figure 1), as has been suggested **by previous studies. Specifically, Pinto et al demonstrated that co-expressed TRGs are co-localised within the same nuclear subdomains via DNA-FISH and Bansal et al. showed that AIRE acts at superenhancers, which are known to localise to the genes they regulate via looping (Bansal et al, 2017; Pinto et al, 2013).**”

We would like to investigate the impact of chromatin looping on TRG co-expression further in the future, but there are currently no TEC genome organisation datasets available in the public domain, and generation of our own dataset lies outside of the scope of this work. Furthermore, a bulk-level Hi-C may not provide the information needed to facilitate this investigation as we would expect changes in chromatin architecture to occur in a small fraction of all mTEC where those TRGs are expressed (TSPAN8+ mTEC ~ 7%, GP2+ mTEC ~ 2%).

Specific questions

1/ Only half of the mTEC in clusters 10 and 14 were positive for the corresponding TRA. Is it due to RNA sampling issues or improper sorting?

We apologise for any confusion, and now have improved the paragraph that was referred to as follows:

“Using FACS to enrich for TSPAN8+ mTEC and GP2+ mTEC, respectively, ensured that we investigated a large number of rare cluster 10 and 14 cells. Nearly half the mTEC in these clusters were positive for their respective TRAs (44% and 49%, respectively) **and the next largest contributor to these clusters were unselected cells for which we have no measurement of TSPAN8 or GP2 protein levels (37% and 39%, respectively).** Importantly, these clusters were robust to clustering of unselected mTEC alone (Figure 3c). Furthermore, while cluster 10 contained thymic tuft cells (Bornstein et al, 2018; Miller et al, 2018), cluster 14 was transcriptionally distinct and expressed a set of chemokine ligands and receptors that are absent from cluster 10.”

The percentages given for TSPAN8 and GP2 positive cells in the paragraph above refer to those positive for protein expression as identified via flow cytometry and not to those positive for mRNA expression (although the two will be related as will be evident by comparing Figures 2d and 3a, b and c, which show the cells' FACS phenotype and hence the protein expression status for TSPAN8 and GP2 with Figures 4e and f, which show the cells' mRNA expression level for the two TRGs). We hope that this is clear from the wording of the paragraph, which refers to FACS enrichment,

uses TSPAN8/GP2 (as opposed to *Tspan8/Gp2*) and TRAs (as opposed to TRGs). The purpose of this paragraph was to highlight that the use of FACS enrichment for rare TRA positive mTEC allowed the investigation of subpopulations of mTEC that express these antigens.

2nd Editorial Decision

22 August 2019

Thank you for submitting the revised manuscript to The EMBO Journal. Your study has now been seen by referee #1 and the comments are provided below. As you can see from the comments, the referee appreciates the introduced changes and are supportive of the study. The referee has a few remaining comments that can be easily resolved with appropriate text changes.

When you re-submit will you also take care of the following points:

- Can you please check the title and if it is clear enough. I find it a bit to fuzzy.
- KW are missing
- COI be changed from "Declaration of competing interests" to "Conflict of interest".
- Fig 3C is not called out.
- Fig 8A is not called out.
- The three tables (Appendix Table S1 etc.) should be renamed "Table EV1" etc. in the files, legends and callouts.
- The appendix file is missing a ToC. Legends for Table S1-S3 should be removed from appendix and added to the tables.
- Our publisher has also done their pre-publication check on your manuscript. When you log into the manuscript submission system you will see the file "Wiley Pre-acceptance check". Please take a look at the word file and the comments regarding the figure legends and respond to the raised issues.
- We also include a synopsis of the paper (see <http://emboj.embopress.org/>). Please provide me with a general summary statement and 3-5 bullet points that capture the key findings of the paper.
- We also need a summary figure for the synopsis. The size should be 550 wide by 400 high (pixels). You can also use something from the figures if that is easier.

That should be all. You can use the revision link below to upload the revised version.

Congratulations on a nice paper!

REFeree REPORTS

Referee #1:

As already stated in the previous summary, the manuscript by Dhalla et al a) validates and strengthens many previous findings regarding TEC heterogeneity and/or promiscuous gene expression, b) brings novel data which dramatically expand TEC heterogeneity (identification of additional TEC subsets that have been overlooked) thanks to very high resolution, c) represents a very valuable resource of data, which will be very instrumental for future studies. Moreover, the authors made a serious effort to address all the points that were previously raised by this reviewer and thereby dramatically improved the clarity of the manuscript, making it suitable for publication in EMBO J.

However, there are still several minor issues, which should be considered, in order to better articulate the novelty, as well as the accuracy of this study

Several minor comments for kind consideration:

1) Abstract:

Although the authors have modified the abstract, it contains some inaccurate/misleading statements and (in my opinion) does not reflect the key novelty of the study. Specifically, the abstract is very focused on whether the process of promiscuous gene expression (PGE) is stochastic or coordinated, while it largely ignores some key and novel findings regarding TEC heterogeneity that were highlighted by this study (i.e. identification of some novel TEC subsets).

The key conclusion of this study (and of the abstract) is that PGE is an ordered process with many stochastic (indeterminate) elements in it. However, a similar conclusion has been reached previously by Meredith et al, who suggested that (Aire-mediated) PGE is neither entirely stochastic nor entirely organized but rather controlled through "organized stochasticity", an ordered process, which depends on stochastic determinism (i.e. chromosome, location, tissue identity, etc of individual genes are indeterminate). In my opinion, the data presented in the paper (and their summary in the abstract) seem to be well in line with this "organized stochasticity" model in spite of some differences in e.g. interindividual variance (which could be explained by different design of both studies)

Moreover, I think the key novelty of this study is that (unlike Meredith et al or other studies) it looks at PGE in all TECs at a single cell level, including Aire-negative TECs. I think this should be better articulated in the abstract and the manuscript, rather than stating that "whether PGE is stochastic or organized is unknown".

Finally, rather than novel insights into PGE, the study, in my opinion, brings several novel and important insights into TEC heterogeneity, as it highlights some previously overlooked TEC subsets including chemokine-expressing or ciliated TECs. Why not stressing these important findings in the abstract and the study itself?

2) Rare clusters 11, 12, 15

Some of the clusters such as fibroblast-like mTEC (12, 15) seem extremely small, raising a question whether they represent a real cluster of a unique TEC subset or a possible artifact (e.g. caused by a rare contamination in which not a single TEC, but rather a doublet of fibroblast and TEC was sequenced?). This should be discussed in the text (e.g. in the discussion, paragraph starting with sentence "We also identified six novel mTEC clusters" may be a good place to discuss that based on their extreme rarity, these may not be bona-fide TEC subsets but contaminants. The other possibility is to validate the actual existence of these rare subsets experimentally.

Finally, when talking about the individual clusters in the text, it would be very useful to indicate how many cells (out of) comprise each cluster.

3) Post-Aire cells

The authors use a term post-Aire mTECs to define clusters 7,8,9. While these are likely post-Aire cells, the labeling may be inaccurate, as a) the authors did not validate this experimentally, b) other clusters may also be post-Aire.

It is maybe better to label them as keratinocyte-like mTECs or Spink5-expressing (or similar) and mention in the text that they likely represent the post-Aire cells described previously.

2nd Revision - authors' response

20 September 2010

Referee #1:

As already stated in the previous summary, the manuscript by Dhalla et al a) validates and strengthens many previous findings regarding TEC heterogeneity and/or promiscuous gene expression, b) brings novel data which dramatically expand TEC heterogeneity (identification of additional TEC subsets that have been overlooked) thanks to very high resolution, c) represents a

very valuable resource of data, which will be very instrumental for future studies. Moreover, the authors made a serious effort to address all the points that were previously raised by this reviewer and thereby dramatically improved the clarity of the manuscript, making it suitable for publication in EMBO J.

However, there are still several minor issues, which should be considered, in order to better articulate the novelty, as well as the accuracy of this study

Several minor comments for kind consideration:

1) Abstract:

Although the authors have modified the abstract, it contains some inaccurate/misleading statements and (in my opinion) does not reflect the key novelty of the study. Specifically, the abstract is very focused on whether the process of promiscuous gene expression (PGE) is stochastic or coordinated, while it largely ignores some key and novel findings regarding TEC heterogeneity that were highlighted by this study (i.e. identification of some novel TEC subsets).

The key conclusion of this study (and of the abstract) is that PGE is an ordered process with many stochastic (indeterminate) elements in it. However, a similar conclusion has been reached previously by Meredith et al, who suggested that (Aire-mediated) PGE is neither entirely stochastic nor entirely organized but rather controlled through "organized stochasticity", an ordered process, which depends on stochastic determinism (i.e. chromosome, location, tissue identity, etc of individual genes are indeterminate). In my opinion, the data presented in the paper (and their summary in the abstract) seem to be well in line with this "organized stochasticity" model in spite of some differences in e.g. interindividual variance (which could be explained by different design of both studies)

Moreover, I think the key novelty of this study is that (unlike Meredith et al or other studies) it looks at PGE in all TECs at a single cell level, including Aire-negative TECs. I think this should be better articulated in the abstract and the manuscript, rather than stating that "whether PGE is stochastic or organized is unknown".

Finally, rather than novel insights into PGE, the study, in my opinion, brings several novel and important insights into TEC heterogeneity, as it highlights some previously overlooked TEC subsets including chemokine-expressing or ciliated TECs. Why not stressing these important findings in the abstract and the study itself?

We have refined the abstract to further highlight the novelty of the study as suggested by Reviewer #1. Accordingly, we now highlight the novel subtypes within the abstract which improves the overall message of the study. The updated abstract is as follows:

“To induce central T-cell tolerance, medullary thymic epithelial cells (mTEC) collectively express most protein coding genes, thereby presenting an extensive library of tissue-restricted antigens (TRAs). To resolve mTEC diversity and whether promiscuous gene expression (PGE) is stochastic or coordinated, we sequenced transcriptomes of 6,894 single mTEC, enriching for 1,795 rare cells expressing either of two TRAs, TSPAN8 or GP2. Transcriptional heterogeneity allowed partitioning of mTEC into 15 reproducible subpopulations representing distinct maturational trajectories, stages and subtypes, including novel mTEC subsets, such as chemokine-expressing and ciliated TEC, which warrant further characterisation. Unexpectedly, 50 modules of genes were robustly defined each showing patterns of co-expression within individual cells, which were mainly not explicable by chromosomal location, biological pathway, or tissue specificity. Further, TSPAN8+ and GP2+ mTEC were randomly dispersed within thymic medullary islands. Consequently, these data support observations that PGE exhibits ordered co-expression, although mechanisms underlying this instruction remain biologically indeterminate. Ordered co-expression and random spatial distribution of a diverse range of TRAs likely enhance their presentation and encounter with passing thymocytes, whilst maintaining mTEC identity.”

2) Rare clusters 11, 12, 15

Some of the clusters such as fibroblast-like mTEC (12, 15) seem extremely small, raising a question whether they represent a real cluster of a unique TEC subset or a possible artifact (e.g. caused by a

rare contamination in which not a single TEC, but rather a doublet of fibroblast and TEC was sequenced?). This should be discussed in the text (e.g. in the discussion, paragraph starting with sentence "We also identified six novel mTEC clusters" may be a good place to discuss that based on their extreme rarity, these may not be bona-fide TEC subsets but contaminants. The other possibility is to validate the actual existence of these rare subsets experimentally.

Finally, when talking about the individual clusters in the text, it would be very useful to indicate how many cells (out of) comprise each cluster.

We thank the reviewer for these comments, and we have updated the relevant Discussion section to address the possibility of contamination or other artefacts:

“The cells in these clusters retained a strong signature of expression of core mTEC genes (gene module 2), and while we cannot rule out contamination by other classes of TEC (cortical TEC) or technical artefacts such as doublets, our repeated observation of these cells across multiple experiments combined with their similarity to other TEC suggested that they represented a rare subpopulation of TEC rather than a contaminant.”

Regarding the number of cells in each cluster, we include the number of cells within each cluster in Figure EV4, which is referenced when appropriate. In our view, the section has improved in its clarity without providing additional information within an already information-dense section of text.

3) Post-Aire cells

The authors use a term post-Aire mTECs to define clusters 7,8,9. While these are likely post-Aire cells, the labeling may be inaccurate, as a) the authors did not validate this experimentally, b) other clusters may also be post-Aire.

It is maybe better to label them as keratinocyte-like mTECs or Spink5-expressing (or similar) and mention in the text that they likely represent the post-Aire cells described previously.

We agree with the reviewer that these cells must be referred to as likely post-Aire cells. The text already lists them as such upon their introduction, but for the sake of simplicity we then dropped the term *likely* for all later references. Upon reflection this perhaps added to confusion so we have accordingly updated all references in the text to include *likely post-AIRE* instead of *post-AIRE*. We believe that this provides greater clarity than the introduction of other terms into the text.

Corresponding Author Name: Chris Ponting
Manuscript Number: EMBOJ-2019-101828